# Network-based systematic dissection of exercise-induced inhibition of myosteatosis in older individuals

Hirotaka Iijima[1,2,3,4,5], Fabrisia Ambrosio[1,2,3] and Yusuke Matsui[5,6]

[1] *Discovery Center for Musculoskeletal Recovery, Schoen Adams Research Institute at Spaulding, Charlestown, MA, USA*
[2] *Department of Physical Medicine & Rehabilitation, Harvard Medical School, Boston, MA, USA*
[3] *Department of Physical Medicine & Rehabilitation, Spaulding Rehabilitation Hospital, Charlestown, MA, USA*
[4] *Institute for Advanced Research, Nagoya University, Nagoya, Japan*
[5] *Biomedical and Health Informatics Unit, Graduate School of Medicine, Nagoya University, Nagoya, Japan*
[6] *Institute for Glyco-core Research, Tokai National Higher Education and Research System, Nagoya University, Nagoya, Japan*

Handling Editors: Paul Greenhaff & Christopher Sundberg

The peer review history is available in the Supporting Information section of this article (https://doi.org/10.1113/JP285349#support-information-section).

**Abstract**  Accumulated fat in skeletal muscle (i.e. myosteatosis), common in sedentary older individuals, compromises skeletal muscle health and function. A mechanistic understanding of how physical activity levels dictate fat accumulation represents a critical step towards establishment of therapies that promote healthy ageing. Using a network medicine paradigm that characterized the transcriptomic response of aged muscle to exercise *versus* immobilization protocols, this study explored the shared molecular cascade that regulates the fate of fibro-adipogenic progenitors (FAPs), the cell population primarily responsible for fat accumulation. Specifically, gene set enrichment analyses with network propagation revealed *Pgc-1α* as a functional hub of a large gene regulatory network underlying the regulation of FAPs by physical activity in aged muscle, but not in young counterparts. Integrated *in silico* and *in situ* approaches to induce *Pgc-1α* overexpression in aged muscle promoted mitochondrial fatty acid oxidation and inhibited FAP adipogenesis. These findings suggest that the *Pgc-1α*–mitochondrial fatty acid oxidation axis is a shared mechanism by which

This article was first published as a preprint. Iijima H, Ambrosio F, Matsui Y. 2023. Pgc-1α is an exercise-responsive regulator of myosteatosis in older individuals. bioRxiv. https://doi.org/10.1101/2023.02.07.527478

physical activity regulates age-related myosteatosis. The network medicine paradigm introduced provides mechanistic insight into exercise adaptation in elderly skeletal muscle and offers translational opportunities to advance exercise prescription for older populations.

(Received 22 July 2023; accepted after revision 10 November 2023; first published online 15 December 2023)

**Corresponding authors** H. Iijima: Discovery Center for Musculoskeletal Recovery, Schoen Adams Research Institute at Spaulding, 149 13th Street, Room 5.106, Charlestown, MA 02129, USA. Email: hiijima1@mgh.harvard.edu; Y. Matsui: Biomedical and Health Informatics Unit, Graduate School of Medicine, Nagoya University, 1-1-20 Daiko-Minami, Higashi-ku, Nagoya, 461-0047, Japan. Email: matsui@met.nagoya-u.ac.jp

**Abstract figure legend** This study introduces a novel network medicine approach, gene set enrichment analysis (GSEA)-guided network propagation, to understand how physical activity regulates fibro-adipogenic progenitor (FAP) adipogenesis in aged skeletal muscle. The culmination of these *in silico* analyses suggests that exercise and immobilization regulate *Pgc-1α* and its downstream target, mitochondrial fatty acid oxidation, to regulate FAP adipogenesis.

## Key points

- Fat accumulation is a quintessential feature of aged skeletal muscle.
- While increasing physical activity levels has been proposed as an effective strategy to reduce the fat in skeletal muscle (i.e. myosteatosis), the molecular cascade underlying these benefits has been poorly defined.
- This study implemented a series of network medicine approaches and uncovered *Pgc-1α* as a mechanistic driver of the regulation of fibro-adipogenic progenitors (FAPs) by physical activity.
- Integrated *in silico* and *in situ* approaches to induce *Pgc-1α* overexpression promoted mitochondrial fatty acid oxidation and inhibited FAP adipogenesis.
- Together, the findings of the current study suggest a novel hypothesis that physical activity reduces myosteatosis via upregulation of *Pgc-1α*-mediated mitochondrial fatty acid oxidation and subsequent inhibition of FAP adipogenesis.

## Introduction

A unique feature of skeletal muscle ageing is accumulation of intra- and intermuscular adipose tissue (IMAT), or myosteatosis, defined as ectopic fat found beneath the fascia and within the muscles (Addison et al., 2014). Even though IMAT accounts for only a small portion of total body fat, it is considered a leading cause of insulin resistance and decreased muscle strength (Biltz et al., 2020; Sachs et al., 2019). IMAT has even been used as a predictor of all-cause and cardiovascular mortality in the elderly (Miljkovic et al., 2015). As such, there is great interest in better understanding the mechanisms of age-related IMAT to guide the development of therapeutic interventions as well as prognostic biomarkers in a geriatric population.

Accumulation of IMAT is driven, at least in part, by muscle-resident fibro-adipogenic progenitors (FAPs), which are identified by expression of platelet-derived growth factor receptor alpha (PDGFRα) (Heredia et al., 2013; Uezumi et al., 2010, 2011). While FAPs are key regulators of muscle regeneration and homeostasis, they also contribute to chronic inflammation, fibrosis and fat deposition when dysregulated (Heredia et al., 2013; Uezumi et al., 2010, 2011). Damage to skeletal muscle triggers a transient phase of FAP proliferation that

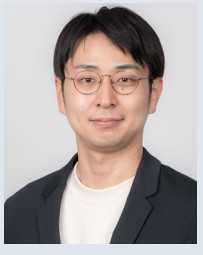

**Hirotaka Iijima** is a research faculty member of the Spaulding Research Institute's Discovery Centre for Musculoskeletal Research and an assistant professor within the Department of Physical Medicine and Rehabilitation at Harvard Medical School. Inspired by previous clinical experience as a physical therapist, he has been working in interdisciplinary fields with the long-term goal of developing innovative rehabilitative strategies for musculoskeletal diseases across the translational spectrum. To achieve this goal, he uses the integrated approaches of rehabilitation medicine together with network medicine, regenerative medicine, and cellular and molecular mechanobiology, under the supervision of Dr Fabrisia Ambrosio at Harvard Medical School. The significance of this line of study is highlighted by the fact that these projects were awarded by the Osteoarthritis Research Society International, the Japanese Society of Cartilage Metabolism, and the Japanese Society for Bone and Mineral Research. He has published over 90 refereed journal articles and reviews in these fields. He currently serves as a guest editor in *JoVE Methods Collections*.

promotes muscle stem cell differentiation, followed by a return to baseline numbers as a result of apoptosis and subsequent phagocytic clearance (Lemos et al., 2015; Mozzetta et al., 2013). However, in the setting of chronic muscle inflammation, as is the case with aged muscle (Wang et al., 2017), FAP proliferation is maintained at high levels owing to impaired apoptosis and removal of dead cells via phagocytosis (Lemos et al., 2015). The result is persistence and adipogenic differentiation of FAPs, thereby inducing the buildup of IMAT (Heredia et al., 2013; Uezumi et al., 2010, 2011).

IMAT accumulation is believed to coincide with a reduced muscle contractile activity. As such, physical exercise has been proposed as a means to counteract an age-related increase in IMAT and promote skeletal muscle health. Indeed, a recent meta-analysis showed that physical exercise reduces IMAT accumulation in elderly individuals with chronic diseases (Tuñón-Suárez et al., 2021). On the other hand, immobilization, defined as restricted muscle contractile activity, increases IMAT (Belavý et al., 2014; Manini et al., 2007). We found only one study demonstrating that physical exercise promotes senescence in FAPs of healthy young skeletal muscle in mice (Saito et al., 2020). As such, the molecular and cellular signalling cascades underlying the exercise- and immobilization-driven regulation of IMAT with ageing remain poorly understood, thereby precluding establishment of effective rehabilitative strategies to improve skeletal muscle health in an elderly population.

As a first step to address this critical knowledge gap, this study applied integrated computational approaches to a publicly available database of the skeletal muscle transcriptomic response and explored the shared mechanism underlying the physical activity-induced IMAT regulation in aged skeletal muscle. Specifically, we implemented a series of gene set enrichment analyses (GSEAs) to the aged skeletal muscle transcriptome and demonstrated that exercise and immobilization differentially regulate FAP adipogenesis. Next, using a series of network medicine-based approaches, this study explored possible mechanistic drivers of FAP adipogenesis. The culmination of these *in silico* analyses suggests that exercise and immobilization regulate *PPARγ coactivator-1α (Pgc-1α)* and its downstream target, mitochondrial fatty acid oxidation, to regulate FAP adipogenesis.

## Materials and methods

### Search for transcriptomic data of elderly skeletal muscle responses to exercise and immobilization

We accessed publicly available transcriptomic data of skeletal muscle generated by the MetaMEx project (Pillon et al., 2020). MetaMEx summarizes skeletal muscle transcriptomic responses to immobilization and exercise (aerobic, resistance and high-intensity training) across the lifespan of human adults (young, middle-aged and elderly) (Pillon et al., 2020). We first accessed all the raw data through GitHub (https://github.com/NicoPillon/MetaMEx). According to the predetermined eligibility criteria, we included studies assessing the transcriptomic response to immobilization or chronic (>3 months) resistance exercise protocols in healthy elderly participants. Elderly was defined as age over 65 years in each study. No restrictions were set according to target muscle of biopsy, quantification method for skeletal muscle transcripts, participants' sex, body mass, race, country or publication year. For the data sets of resistance exercise, this study limited participants with physical activity level of sedentary because of possible confounding effects on the transcriptomic response to the exercise protocol (Rubenstein et al., 2022). The MetaMEx database search was performed in November 2022 by a single reviewer (H.I.). During these processes, the reviewer prepared and used simple, predesigned Google spreadsheets to assess eligibility by extracting study features.

### Identification of genes associated with insulin sensitivity in elderly skeletal muscle

Genes associated with insulin sensitivity were defined according to analysed data in Lutter et al. (2022). The original study published by Lutter et al. (2022) obtained transcriptomic profile data as well as clinical data of insulin sensitivity (e.g. glucose infusion rate) from skeletal muscle and IMAT in 16 elderly participants with obesity and type II diabetes mellitus from a cross-sectional study previously published by Sachs et al. (2019). After multivariate regression analysis in which gene expression was regressed to insulin sensitivity, Lutter et al. (2022) identified 59 genes positively associated with good insulin sensitivity. We accessed this list of genes and used it for further analyses.

### Target gene prediction of miR-206

Primary target genes of differentially expressed micro-RNAs (miRNAs) were determined using miRTarBase (Huang et al., 2020). miRTarBase is an experimentally validated miRNA–target interactions database (Huang et al., 2020). Among the experimentally validated miRNAs, we have included genes with 'strong' experimental evidence (i.e. genes supported by reporter assay, western blot or qRT-PCR).

### Mouse model of skeletal muscle-specific overexpression of *Pgc-1α*

We accessed archived RNA-sequencing (RNA-seq) data collected from quadriceps muscle in older–old

(29–34 months old) male and female C57BL/6J mice (Garcia et al., 2018). Their original study identified 1485 differentially regulated genes (false discovery rate <0.05, fold change >2) across muscle-specific *Pgc-1α* over-expression mice ($n = 3$) *versus* wild-type mice ($n = 4$), of which 885 and 600 were up- and downregulated, respectively (Garcia et al., 2018). Since the transcriptomic response was similar across males and females, the RNA-seq data from males and females were mixed and used for further analyses.

## RNA-seq analysis to define genes associated with FAP adipogenesis

We accessed the archived RNA-seq data (GSE171502) collected from isolated FAPs with and without miR-206 deficiency [as induced by small interferring (siRNA) to miR-206] (Wosczyna et al., 2021). The miR-206 deficiency recapitulated increased adipogenic differentiation of PDGFRα-positive FAPs *in vitro* (Wosczyna et al., 2021). Raw count data were normalized by counts per million (CPM), filterByExpr function and Trimmed Mean Mvalue (TMM) using the R/Bioconductor package edgeR with default parameters (Robinson et al., 2010). Differential gene expression analysis was performed for genes with a normalized CPM value using the R/Bioconductor package limma (Ritchie et al., 2015). The Benjamini–Hochberg false discovery rate (FDR) control for multiple hypothesis testing was used to produce $q$-values.

## Filtering low-expression genes in MetaMEx transcriptomic data

We removed genes that have very low counts in RNA-seq data or signal intensity in microarray data prior to downstream analysis on biological and statistical grounds (Chen et al., 2016). From a biological point of view, a gene must be expressed at some minimal level before it is likely to be translated into a protein or to be considered biologically important. From a statistical point of view, genes with consistently low counts or low signal intensity are very unlikely be assessed as differentially expressed because low counts or low signal intensity do not provide enough statistical evidence for a reliable judgement to be made. Such genes can therefore be removed from the analysis without any loss of information. We used the filterByExpr function with default parameters for low count data (min count = 10) in RNA-seq data (Chen et al., 2016). For signal intensity in microarray data, we used a soft intensity-based filter (Klaus & Reisenauer, 2016) as recommended by the R/Bioconductor package limma user guide (Ritchie et al., 2015).

## Functional characterization of the transcriptome using pathway enrichment analysis

To determine the biological function of genes of interests, Wiki pathway enrichment analysis was performed by Enrichr software (Kuleshov et al., 2016). WikiPathways (wikipathways.org) captures the collective knowledge represented in biological pathways (Slenter et al., 2018). WikiPathways provides easy-to-use drawing and annotation tools to capture identities, relationships, comments and literature references for each pathway element and interaction.

## Gene set enrichment analysis

GSEA was performed as described previously (Subramanian et al., 2005) using the GSEA web tool provided by the Broad Institute (https://www.gsea-msigdb.org/gsea/index.jsp). This study implemented single sample GSEA (ssGSEA), an extension of GSEA that allows one to define an enrichment score that represents the degree of absolute enrichment of a gene set in each sample within a given data set (Barbie et al., 2009). The transcriptomic responses (i.e. log fold change) of elderly skeletal muscle to resistance exercise and immobilization relative to baseline control were rank-normalized and used for input. The genes associated with insulin sensitivity or FAP adipogenesis we originally defined were used as a gene set. The minimum and maximum criteria for selection of gene sets from the collection were 10 and 500 genes, respectively.

Leading edge analysis was performed after each GSEA to determine the core genes defining the subset of genes with positive contribution to the enrichment score before it reaches its peak; that is, those that are most correlated with the phenotype of interest. These leading edge genes were used as seeded genes for network propagation with gene set enrichment analysis (random-walk-restart, RWR).

## WGCNA and hub gene analysis

This study used the Weighted gene co-expression network analysis (WGCNA) package to build a weighted gene co-expression network using the archived RNA-seq or microarray data in the MetaMEx database, which finally yielded 7468 genes identified across the different data sets after filtering low expression genes, as described above. The key parameter, $\beta$, for weighted network construction was optimized to maintain both the scale-free topology and sufficient node connectivity as recommended in the manual. A topological

overlap matrix (TOM) was then formulated based on the adjacency values to calculate the corresponding dissimilarity (1 − TOM) values. Module identification was accomplished with the dynamic tree cut method by hierarchically clustering genes using 1 − TOM as the distance measure with a minimum size cutoff of 30 and a deep split value of 2 for the resulting dendrogram. A module preservation function was used to verify the stability of the identified modules by calculating module preservation and quality statistics in the WGCNA package (Langfelder & Horvath, 2008).

From the modules constructed by WGCNA, we identified hub gene via Cytoscape software (version 3.9.1) using the 'analyse network' option which determines the topological parameters (e.g. EdgeCount and NeighborhoodConnectivity) of given networks. In the provided parameters, 'EdgeCount' describes the link between one node with its adjacent nodes and the node with the highest 'EdgeCount' was defined as the hub gene in the network.

### Network propagation using RWR

RWR was performed by using the R/Bioconductor package RandomWalkRestartMH (Valdeolivas et al., 2019). RWR simulated a walker starting from one node or a set of nodes (seed nodes) in one network, and such walker randomly moved in the network to deliver probabilities on the seed nodes to other nodes. After iteratively reaching stability, the affinity score of all nodes in the given network to the seeded node was obtained.

### Quantification of percentage rejuvenation of *Pgc-1α* induced by resistance exercise

We accessed two data sets (GSE28422 and GSE97084) (Raue et al., 2012; Robinson et al., 2017) of skeletal muscle in (1) healthy young without exercise intervention (at baseline), (2) elderly without exercise intervention (at baseline) and (3) elderly with resistance exercise. From *Pgc-1α* gene expression data across the three conditions, we calculated the percentage rejuvenation of *Pgc-1α* expression (0%: elderly skeletal muscle without exercise, 100%: healthy young muscle without exercise).

### Unsupervised machine learning

Principal components analysis (PCA) was performed for data reduction to identify the principal components that represent differences in the given data sets using JMP Pro 16 software (SAS Institute, Cary, NC, USA). PCA produces linear combinations of the original variables to generate the principal components (Wold et al., 1987), and visualization is generated by projecting the

original data to the first two principal components. Prior to analysis, we checked assumptions for linearity and suitability of data reduction via a correlogram, as per Bair et al. (2006). To further confirm the linearity, a Pearson correlation analysis was performed for randomly selected genes, which supported the linear relationship of the transcriptomic response among genes. In addition, we confirmed the significance of data reduction *via* a PCA scree plot, which represents the amount of variance explained by each principal component. To assess the robustness of PCA to small sample size, we implemented a leave-one-out algorithm (i.e. one randomly selected data set and repeated for all data sets). We confirmed similar PCA results to the original analysis, as documented by a clear segregation in the transcriptomic response when comparing the exercise and immobilization protocols.

### Statistical analysis

All statistical analyses were performed using JMP Pro 16 software. Except where indicated, data are displayed as means, with uncertainty expressed as 95% confidence intervals (mean ± 95% CI). Due to the exploratory nature of the study, the sample size was not pre-determined. A Mann–Whitney $U$ test or linear regression analysis were performed. In the linear regression analysis, we checked (1) the normality of residuals using a Shapiro–Wilk test, and (2) the homogeneity of the variances as well as linearity by comparing the residuals *versus* fitted values (i.e. the residuals had to be normally distributed around zero). No correction was applied for multiple comparison because outcomes were determined a priori and were highly correlated. No statistical analyses included confounders (e.g. mean body mass in each study or data set) due to the small sample size. We conducted a complete-case analysis in the case of missing data for RNA-seq and/or microarray (i.e. genes under detection). As the missing data with low counts or low signal intensity do not provide enough statistical evidence for a reliable judgement to be made, we did not consider multiple imputation. In all experiments, $P$-values $<0.05$ were considered statistically significant. Throughout this text, '$n$' represents the number of independent observations of the study or data set. Specific data representation details and statistical procedures are also indicated in the figure legends.

## Results

### Resistance exercise and immobilization differentially regulated FAP adipogenesis genes

While accumulated evidence has shown that IMAT is affected by muscle contractile activity (i.e. IMAT

is reduced by resistance exercise but increased by immobilization) (Belavý et al., 2014; Manini et al., 2007; Tuñón-Suárez et al., 2021), the underlying molecular mechanism is unclear. To address this gap, we first summarized the transcriptomic response of elderly skeletal muscle to resistance exercise *versus* immobilization. For this, we accessed MetaMEx, a public database of transcriptomic responses to an array of exercise and immobilization protocols in human skeletal muscle (Pillon et al., 2020). From 50 studies with 83 data sets curated through the MetaMEx database, our current study finally included four studies with seven data sets for chronic resistance exercise (three studies with five data sets; mean age: 75.9 years old; body mass index: 25.1 kg/m$^2$; $n = 36$ participants) (Hangelbroek et al., 2016; Raue et al., 2012; Robinson et al., 2017) and immobilization (one study with two data sets; mean age: 68.0 years old; body mass index: 25.1 kg/m$^2$; $n = 18$ participants) (Mahmassani et al., 2019) in older individuals (Fig. 1*A* and *B*). Studies with non-healthy adults (e.g. with diabetes and/or metabolic syndrome), young or middle-aged adults, non-resistance exercise (e.g. aerobic exercise), adults who performed life-long activity and adults who participated in athlete-level physical activity were excluded to minimize possible confounding effects on the transcriptomic response to resistance exercise *versus* immobilization protocols.

In the included studies, healthy elderly participants either completed a resistance exercise protocol 3 days/week for 3−6 months or were immobilized through bed rest for a total of 5 days. All participants in the resistance exercise group had sedentary behaviours at baseline and reported not participating in any formal exercise programmes or regular physical activity when enrolling in the study (Hangelbroek et al., 2016; Raue et al., 2012; Robinson et al., 2017). Similarly, all participants in the immobilization group had minimal levels of physical activity at baseline (Mahmassani et al., 2019). In these studies, skeletal muscle tissue was collected from vastus lateralis muscle and used for RNA-seq (Illumina HiSeq) or microarray (Affymetrix) analyses. The integrated transcriptomic data from four studies with seven data sets included 13,589 genes. To account for the bias caused by low expression count data, we applied a filter based on the gene expression proposed by Chen et al. (2016) to the original RNA-seq data [GSE97084 (Robinson et al., 2017) and GSE113165 (Mahmassani et al., 2019)]. Similarly, to account for the bias caused by low signal intensity data, we applied filtering based on signal intensity (Ritchie et al., 2015) to the original microarray data [GSE28422 (Raue et al., 2012) and GSE117525 (Hangelbroek et al., 2016)]. These two different filters finally resulted in 7468 genes that were used for further analyses.

We first investigated whether resistance exercise and/or immobilization changes the expression of genes associated with insulin resistance in elderly skeletal muscle, a primary adverse clinical consequence directly caused by IMAT accumulation (Sachs et al., 2019). We accessed a list of 59 genes associated with glucose infusion rate of skeletal muscle (i.e. insulin sensitivity) and/or IMAT in the elderly with obesity and type II diabetes mellitus (Lutter et al., 2022) (Fig. 1*C*). PCA revealed a clear segregation in the transcriptomic response of these insulin sensitivity-related genes when comparing immobilization and resistance exercise protocols (Fig. 1*D*). Further, the principal component score, which represents overall gene expression of the insulin sensitivity genes, was different across resistance exercise and immobilization groups (Fig. 1*E*). Since the 59 genes were positively associated with glucose infusion rate (i.e. greater insulin sensitivity) (Lutter et al., 2022), we interpreted this finding as suggesting that resistance exercise improved, while immobilization worsened, insulin sensitivity. We have also provided the data from healthy young adults as a reference value (exercise: GSE28422 and GSE97084; immobilization: GSE113165) and revealed the comparable transcriptomic response of insulin sensitivity across elderly and healthy young muscle (Fig. 1*D* and *E*). Opposing effects of resistance exercise and immobilization on insulin sensitivity are well known clinically (Booth, 1982; Goodyear & Kahn, 1998), thereby indicating the utility of the transcriptomic profile as a reflection of clinical endpoints. Of note, even when we considered all 7468 genes, resistance exercise and immobilization induced distinct transcriptomic responses (Fig. 2*A* and *B*) and were found to exert opposing effects on metabolism-related pathways, such as 'Electron transport chain' and 'Mitochondrial complex I assembly model OXPHOS system' (Fig. 2*C*). These findings are consistent with the above results that included only the insulin sensitivity genes (Fig. 1*C*–*E*).

Over the last decade, numerous studies have provided important insights into the cellular origin of IMAT (Heredia et al., 2013; Uezumi et al., 2010, 2011). Skeletal muscle contains FAPs that do not form adipocytes under conditions of homeostasis. However, in response to injury, FAPs proliferate and support the commitment of myogenic progenitor cells during muscle repair (Joe et al., 2010). The balance between FAPs and myogenic progenitors is, however, disrupted in aged skeletal muscle, owing in large part to chronic inflammation (Franceschi et al., 2018), which drives FAPs to differentiate into adipocytes and give rise to IMAT (Gorski et al., 2018; Joe et al., 2010; Uezumi et al., 2010). Therefore, we next investigated whether exercise and immobilization differentially regulate genes contributing to the adipogenic differentiation of FAPs in aged skeletal muscle. Although no study to our knowledge has identified genes that drive FAP adipogenic differentiation, Wosczyna et al. (2021) showed that miR-206 mimicry *in vivo* limits IMAT, while miR-206 inhibition by siRNA increased the adipogenic

differentiation of FAPs *in vitro*. Given that these previous experiments were designed to elucidate the fate switch from FAPs to adipocytes, the investigators evaluated gene expression *before* mature adipocytes were formed (Wosczyna et al., 2021).

A previous study showed that an acute bout of exercise increases pri-miR-206, the initial product of miR-206 gene transcription, in elderly skeletal muscle (Drummond et al., 2008). These results raised the novel hypothesis that exercise counteracts the FAP adipogenesis programme

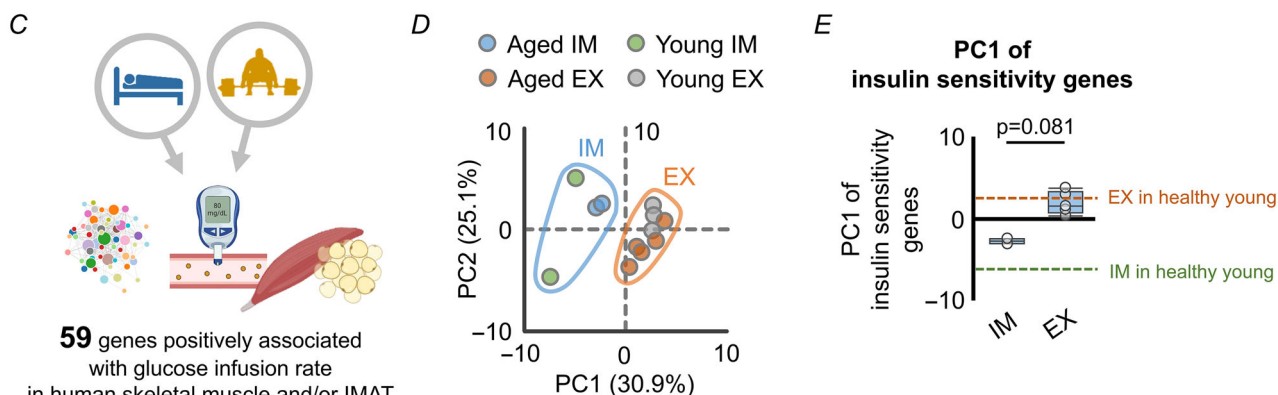

**Figure 1. Transcriptomic response of insulin sensitivity-related genes to resistance exercise and immobilization protocols**

*A*, four studies with seven data sets (Hangelbroek et al., 2016; Koh et al., 2022; Mahmassani et al., 2019; Raue et al., 2012) from the MetaMEx database (Pillon et al., 2020) were included for *in silico* analyses investigating the transcriptomic response of aged skeletal muscle to chronic (≥3 months) resistance exercise (three studies with five data sets) or immobilization (one study with two data sets). *B*, participants' characteristics. Each dot represents a single data set. *C*, schematic showing the concept of the analysis to characterize the transcriptomic response of 59 genes positively associated with glucose infusion rate (i.e. insulin sensitivity) in elderly skeletal muscle. The list of 59 genes was collected from previous literature (Lutter et al., 2022). *D*, PCA showing the separate clusters in immobilization (blue area; young: $n = 2$, aged: $n = 2$) and resistance exercise (orange area; young: $n = 3$, aged: $n = 5$) against overall expression of 59 genes positively associated with glucose infusion rate in elderly skeletal muscle. *E*, PC1 extracted from the PCA revealed a distinct transcriptomic response of the insulin sensitivity-related genes to immobilization ($n = 2$) and resistance exercise protocols ($n = 5$) in elderly individuals. Healthy young values are also provided for visualization purpose (dotted line; exercise: $n = 3$, immobilization: $n = 2$). Statistical analysis was performed using a Mann–Whitney $U$ test (*E*). Parts of the figures were created with biorender.com. Abbreviations: EX, exercise; IM, immobilization; IMAT: intra- and intermuscular adipose tissue; PC, principal component; PCA, principal component analysis. [Colour figure can be viewed at wileyonlinelibrary.com]

that is triggered by knockout of miR-206. To test this hypothesis, we accessed the archived RNA-seq data (GSE171502) that collected RNA from isolated FAPs with and without siRNA to miR-206 (Woszczyna et al., 2021) (Fig. 3*A*). Our RNA-seq analysis identified 161 differentially expressed genes (FDR < 0.05) that may drive adipocytic differentiation of FAPs (Fig. 3*A*). Indeed, these genes were significantly enriched to adipogenesis-related pathways such as 'Adipogenesis' and 'PPAR signalling' (Fig. 3*B*). Using these 161 genes, we then performed ssGSEA (Barbie et al., 2009) to determine whether resistance exercise and immobilization regulate FAP adipogenesis in opposing directions, as we observed above for insulin sensitivity-related genes (Fig. 1*C–E*). GSEA is a computational tool that provides insight into biological processes or pathways underlying a given phenotype (Sub-

ramanian et al., 2005). An extension of GSEA (Barbie et al., 2009), ssGSEA, calculates separate enrichment scores for each paring of a sample and gene set. Results from the available resistance exercise data sets revealed significant changes in FAP adipogenesis genes, while immobilization significantly changed FAP adipogenesis genes in the opposite direction (Fig. 3*C*). We also analysed male and female data separately to consider sex as a biological variable (Ansdell et al., 2020), and found that male and female participants displayed similar transcriptomic responses to both resistance exercise and immobilization (Fig. 3*C*). Intriguingly, the impact of resistance exercise and immobilization on FAP adipogenesis had minimal effect on FAP adipogenesis in healthy young skeletal muscle (Fig. 3*D*). These results indicate that resistance exercise and immobilization differentially regulate

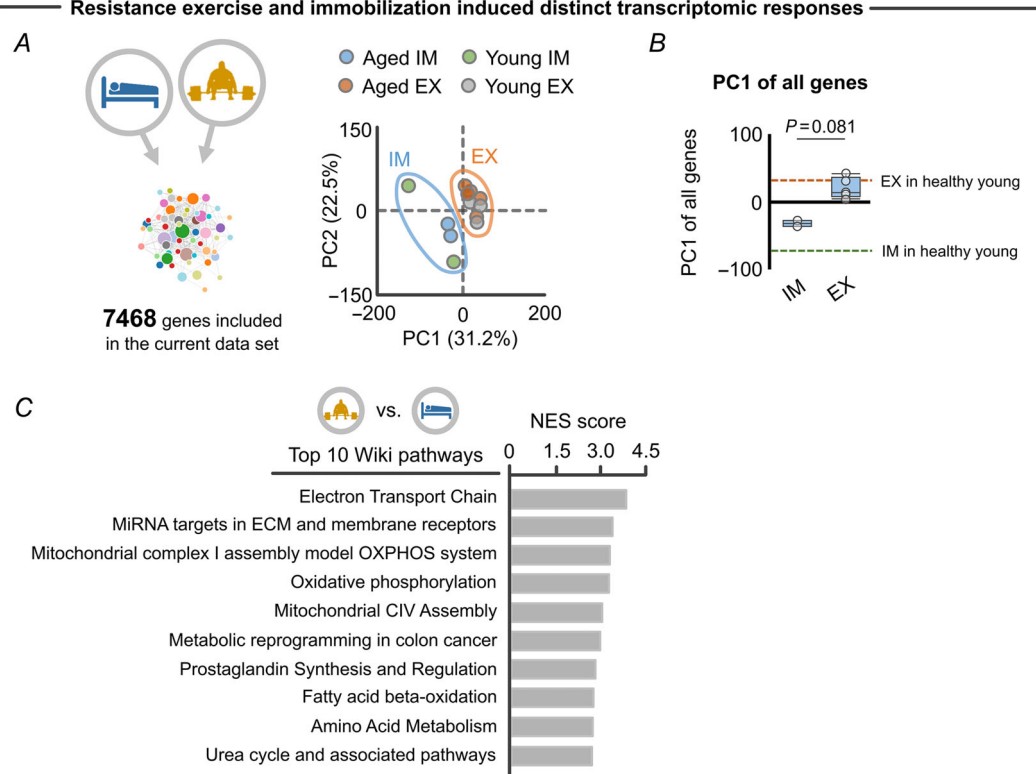

**Figure 2. Overall transcriptomic response of 7468 genes to resistance exercise and immobilization protocols in elderly individuals**

*A*, PCA showing the separate clusters in immobilization (blue area; young: *n* = 2, aged: *n* = 2) and resistance exercise (orange area; young: *n* = 3, aged: *n* = 5) against overall expression of 7468 genes included in the archived transcriptomic data. *B*, PC1 extracted from the PCA revealed a distinct transcriptomic response of 7468 genes to immobilization (*n* = 2) and resistance exercise protocols (*n* = 5) in elderly individuals. Healthy young values are also provided for visualization purposes (dotted line; exercise: *n* = 3, immobilization: *n* = 2). *C*, top 10 Wiki pathways significantly (FDR < 0.05) enriched to PC1 score (i.e. score that differentiate transcriptomic response of resistance exercise and immobilization protocols) of 7468 genes. In this analysis, GSEA was performed to calculate NES and FDR, in which the loading score of each gene contributing to the PC1 score was included as an input. Wiki pathways are in order according to NES score for visualization purpose. Statistical analysis was performed using a Mann–Whitney *U* test (*B*). Portions of the figures were created with biorender.com. Abbreviations: ECM, extracellular matrix; EX, exercise; FDR, false discovery rate; GSEA, gene set enrichment analysis; IM, immobilization; NES, normalized enrichment score; PC, principal component; PCA, principal component analysis. [Colour figure can be viewed at wileyonlinelibrary.com]

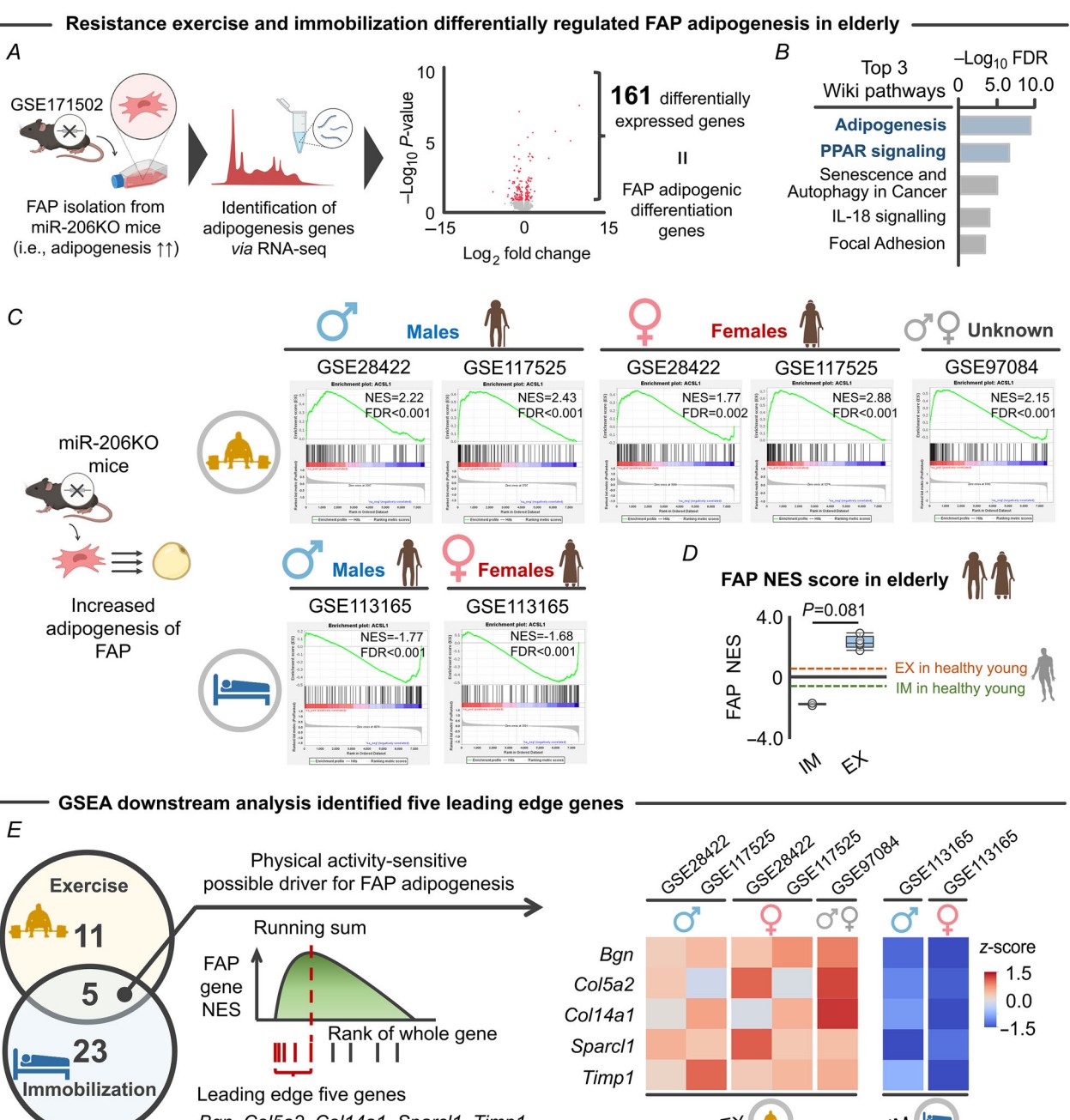

**Figure 3. Transcriptomic response of FAP adipogenesis genes to resistance exercise and immobilization protocols**

*A*, schematic showing the process of defining the genes associated with FAP adipogenic differentiation. From archived RNA-seq of FAPs isolated from the skeletal muscle of wild-type and miR-206 KO mice (GSE113165; Mahmassani et al., 2019), 161 differentially expressed genes associated with FAP adipogenesis were identified. *B*, top five Wiki pathways significantly enriched to 161 differentially expressed genes associated with FAP adipogenesis. *C*, single sample GSEA revealed that resistance exercise and immobilization regulated FAP adipogenesis genes in opposing directions, regardless of sex. *D*, impact of exercise and immobilization on FAP adipogenesis in elderly skeletal muscle was higher than in young counterparts (dotted line; exercise: *n* = 3, immobilization: *n* = 2). *E*, leading edge analysis identified five genes (*Bgn*, *Col5a2*, *Col14a1*, *Sparcl1* and *Timp1*) that overlapped across exercise and immobilization protocols. Numbers in the Venn diagram represent the number of leading edge genes. Colour in the heatmap indicates the *z*-score for each gene. Portions of the figures were created with biorender.com. Statistical analyses were performed using a Mann–Whitney *U* test (*D*). Abbreviations: EX, exercise; FAPs, fibro-adipogenic progenitors; FDR, false discovery rate; GSEA, gene set enrichment analysis; IM, immobilization; KO, knock out; NES, normalized enrichment score. [Colour figure can be viewed at wileyonlinelibrary.com]

miR-206-dependent FAP adipogenesis in elderly individuals, independent of sex.

To further elaborate the observed relationship between FAP adipogenesis genes and physical activity levels (i.e. exercise *vs.* immobilization), we next performed leading edge analysis of GSEA, which allows for determination of subsets of genes that contributed the most to the enrichment signal of a given gene set (referred to as the 'leading edge subset') (Subramanian et al., 2005). Leading edge analysis is done from the enrichment score, which is defined as the maximum deviation from zero (Subramanian et al., 2005). Genes that comprise a leading edge subset have a high correlation between their expression level and the phenotype in question (i.e. FAP adipogenesis) and tend to be at the extremes of the distribution, rather than randomly distributed (Subramanian et al., 2005). We found that five leading edge genes (*Bgn*, *Col5a2*, *Col14a1*, *Sparcl1* and *Timp1*) were significantly changed across the exercise and immobilization protocols, in which resistance exercise upregulated these genes while immobilization down-

regulated their expression (Fig. 3*E*). Among the identified leading edge genes, *Sparcl1*, a member of the SPARC family, is of particular interest given a recent study demonstrating that *Sparcl1* regulates adipogenesis in mice (Meissburger et al., 2016). Taken together, the findings suggest that the FAP adipogenic programme is suppressed by resistance exercise but exacerbated by immobilization.

To further support the evidence regarding the impact of exercise and immobilization on miR-206-dependent FAP adipogenesis, we conducted an additional analysis using the same MetaMEx data sets. Our aim was to determine whether chronic resistance exercise and immobilization induce distinct alterations in the expression level of miR-206 target genes. According to target gene prediction via miRTarBase (Huang et al., 2020), we identified 50 genes that are directly targeted by miR-206 (Fig. 4*A*). Consistent with FAP adipogenesis genes (Fig. 3), the exercise and immobilization protocols differentially regulated miR-206 target genes, as assessed by ssGSEA (Fig. 4*B*) and PCA (Fig. 4*C* and *D*).

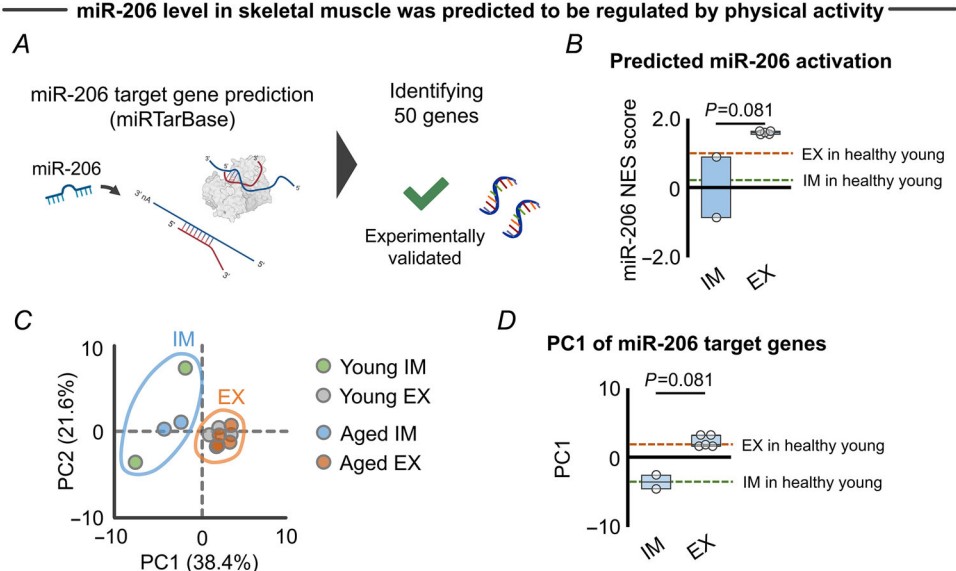

**Figure 4. miR-206 enrichment analysis for transcriptomic response to resistance exercise and immobilization protocols in elderly individuals**

*A*, target gene of miR-206 was determined via miRTarBase (Huang et al., 2020), which identified 50 target genes supported by strong evidence (i.e. target genes have been validated by reporter assay, western blot or qPCR). *B*, predicted miR-206 activation level for the transcriptomic response to immobilization ($n = 2$) and resistance exercise protocols ($n = 5$) in elderly individuals. miR-206 enrichment analysis was performed via single sample GSEA, in which miR-206 target genes were included as a gene set. Healthy young values were also provided for visualization purposes (dotted line; exercise: $n = 3$, immobilization: $n = 2$). *C*, PCA showing the separate clusters in immobilization (blue area; young: $n = 2$, aged: $n = 2$) and resistance exercise (orange area; young: $n = 3$, aged: $n = 5$) against overall expression of 50 miR-206 target genes. *D*, PC1 extracted from the PCA revealed a distinct transcriptomic response of miR-206 target genes to immobilization ($n = 2$) and resistance exercise protocols ($n = 5$) in elderly individuals. Healthy young values are also provided for visualization purposes (dotted line; exercise: $n = 3$, immobilization: $n = 2$). Portions of the figures were created with biorender.com. Statistical analyses were performed using a Mann–Whitney $U$ test (*B*, *D*). Abbreviations: EX, exercise; GSEA, gene set enrichment analysis; IM, immobilization; NES, normalized enrichment score; PC, principal component; PCA, principal component analysis. [Colour figure can be viewed at wileyonlinelibrary.com]

## WGCNA identified a *Pgc-1α*-centredcene module associated with FAP adipogenesis genes that are regulated by exercise and immobilization

We next sought to identify mechanistic drivers of FAP adipogenesis that are differentially regulated by resistance exercise and immobilization. Of the five identified genes, only *Sparcl1* has been investigated in the context of skeletal muscle adipogenesis, while the role of the other four leading edge genes (*Bgn*, *Col5a2*, *Col14a1* and *Timp1*) remains unexplored. As such, our interest was directed to other genes that have similar functions with the identified leading edge genes with the aim of obtaining further mechanistic insight into how physical activity levels regulate FAP adipogenesis. Network medicine-based evidence states that when a gene or molecule is implicated as having a pathogenic role, its direct interactors are likely to play a role in the same pathological process (Barabási et al., 2011). According to this 'disease module hypothesis', possible mechanistic drivers of exercise- and immobilization-induced FAP adipogenesis are likely to be located in the neighbourhood (or module) of the five leading edge genes in the functional network. With this in mind, we first constructed a data-driven regulatory gene network based on the transcriptomic data from aged skeletal muscle (Fig. 5*A*). WGCNA is a data-driven approach to generate gene–gene co-expression networks from all pairwise correlations (Langfelder & Horvath, 2008; Zhang & Horvath, 2005), allowing for the identification of groups or modules of genes that are highly co-expressed and functionally related (Langfelder & Horvath, 2008; Zhang & Horvath, 2005). In this study, data of transcriptomic responses (log-fold change values) after exercise or immobilization protocols relative to the respective baseline controls were used as input. While the network constructed by WGCNA is commonly referred to as a 'co-expression network', given the responsive nature of the network in this study (i.e. the network represents the molecular response to immobilization and exercise), we refer to it as a 'responsive gene regulatory network' in the following analyses. Using the responsive gene regulatory network, we posited that the mechanistic driver (e.g. transcriptional regulator) of FAP adipogenesis is in the vicinity of the five leading edge genes we identified (Fig. 5*B*).

Intriguingly, four (*Bgn*, *Col14a1*, *Sparcl1* and *Timp1*) of the five leading edge genes identified by GSEA were included in module 2, the second largest module (1137 genes) among the 39 different modules constructed (Fig. 5*C*). FAP adipogenesis genes were similarly highly enriched in module 2 (Fig. 5*D*). Eigengene expression (a first principal component of a given module) of module 2 was further different across the immobilization and exercise protocols (Fig. 5*E*). These findings indicate that genes in module 2 represent FAP adipogenesis genes that

are sensitive to resistance exercise and immobilization, suggesting that module 2 may include mechanistic drivers of FAP adipogenesis. Indeed, Wiki pathway enrichment analysis (Slenter et al., 2018) identified pathways related to muscle contractile activity and adipogenesis, such as the 'Striated muscle contraction pathway', 'Fatty acid beta-oxidation' and 'Mitochondrial LC-fatty acid beta-oxidation' (Fig. 5*F*). Of note, genes in module 8, another module that included one leading edge gene (*Col5a2*), were predominantly associated with 'Focal adhesion', but not adipogenesis-related pathways. The distinct pathway enrichment of modules 2 and 8 implies a unique biological role of module 2 in regulating adipogenesis in skeletal muscle. Most notably, the constructed responsive gene regulatory network identified *Pgc-1α* as a hub gene in module 2 (Fig. 5*G*). *Pgc-1α* is a transcriptional co-activator known to play a major role in adaptive remodelling of skeletal muscle mitochondria (Garcia et al., 2018; Koves et al., 2005). *Pgc-1α* also targets a number of genes involved in lipid droplet assembly and mobilization in response to exercise (Koves et al., 2013). Building upon this previous work, the present study identified a large gene module that represents biological function associated with FAP adipogenesis, in which *Pgc-1α* was implicated as a central regulator.

## GSEA-guided network propagation identified *Pgc-1α* as a possible driver of FAP adipogenesis regulated by exercise and immobilization

To further support the finding that *Pgc-1α* is a hub gene in module 2 and therefore a possible mechanistic driver of the regulatory effects of physical activity levels on FAP adipogenesis, we here introduced GSEA-guided network propagation on the responsive gene regulatory network. Network propagation explores the network vicinity of genes seeded to study their biological functions based on the premise that nodes related to similar functions tend to lie close to each other in the networks (Cowen et al., 2017) (Fig. 6*A*). We used an RWR algorithm (Valdeolivas et al., 2019) to verify which nodes are more frequently visited on a random path in a given network. We posited that RWR starting at the five leading edge genes (i.e. *in silico* over-expression) would consistently pseudo-activate (i.e. affinity score >0) the *Pgc-1α* gene node (Fig. 6*B*). Figure 6*C* shows an example of *in silico* over-expression of *Sparcl1* on the network.

As hypothesized, *in silico* pseudo-activation of the five genes consistently pseudo-activated the *Pgc-1α* gene node (Fig. 6*D*). Of note, the responsive gene regulatory network was constructed using the top 5000 co-regulated gene pairs as a default. To address the possibility that different numbers of co-regulated gene pairs for the

**Leading edge genes were clustered in the same regulatory network constructed by WGCNA**

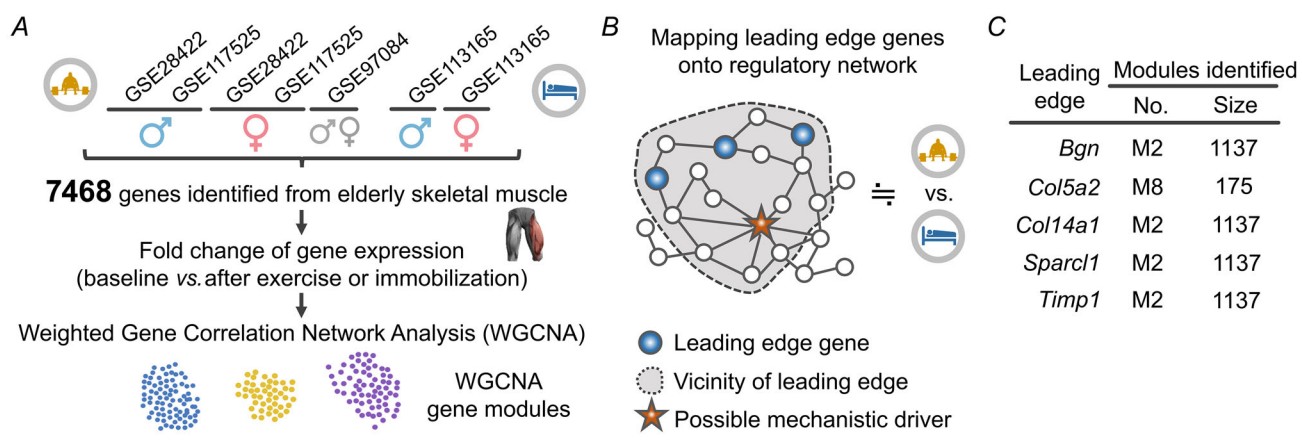

**WGCNA identified *Pgc-1α*-containing module with high enrichment to FAP adipogenesis genes**

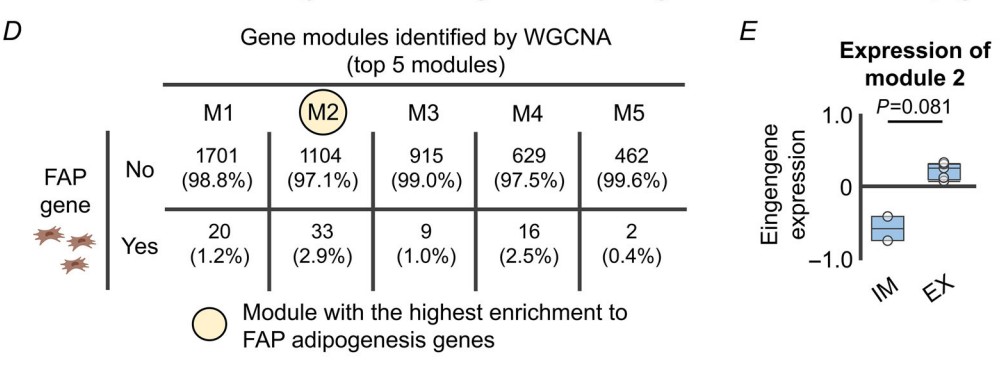

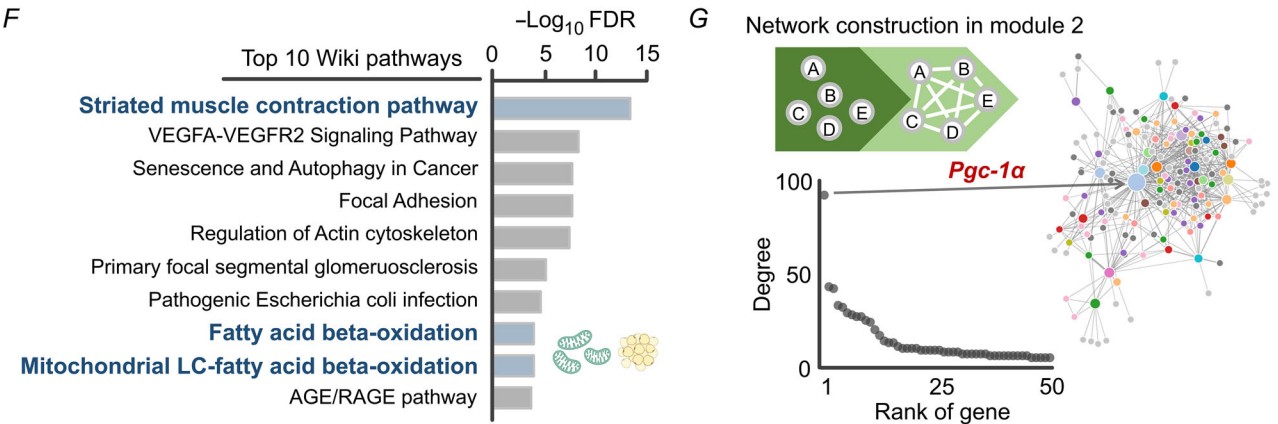

**Figure 5. WGCNA revealed co-regulatory relationship between *Pgc-1α* and FAP adipogenesis**
*A*, schematic showing the analytical flow for WGCNA from the archived transcriptomic data (7468 genes) (Hangelbroek et al., 2016; Koh et al., 2022; Mahmassani et al., 2019; Raue et al., 2012). *B*, schematic showing the concept of identification of a mechanistic driver in a regulatory network constructed by WGCNA under the assumption of the disease module hypothesis. *C*, four of the five leading edge genes were included in module 2, the second largest module constructed. *D*, among the 39 modules, module 2 displayed the highest enrichment to FAP adipogenesis genes. Values in the 2 × 5 contingency table show number of genes (percentage). *E*, eigengene expression of module 2 was different between the immobilization (*n* = 2) and exercise protocols (*n* = 5). *F*, top 10 Wiki pathways significantly enriched to module 2. *G*, responsive gene regulatory network of module 2. *Pgc-1α* was identified as a hub gene. Statistical analyses were performed using a Mann–Whitney *U* test (*E*). Abbreviations: EX, exercise; FAPs, fibro-adipogenic progenitors; FDR, false discovery rate; IM, immobilization; WGCNA, weighted gene correlation network analysis. [Colour figure can be viewed at wileyonlinelibrary.com]

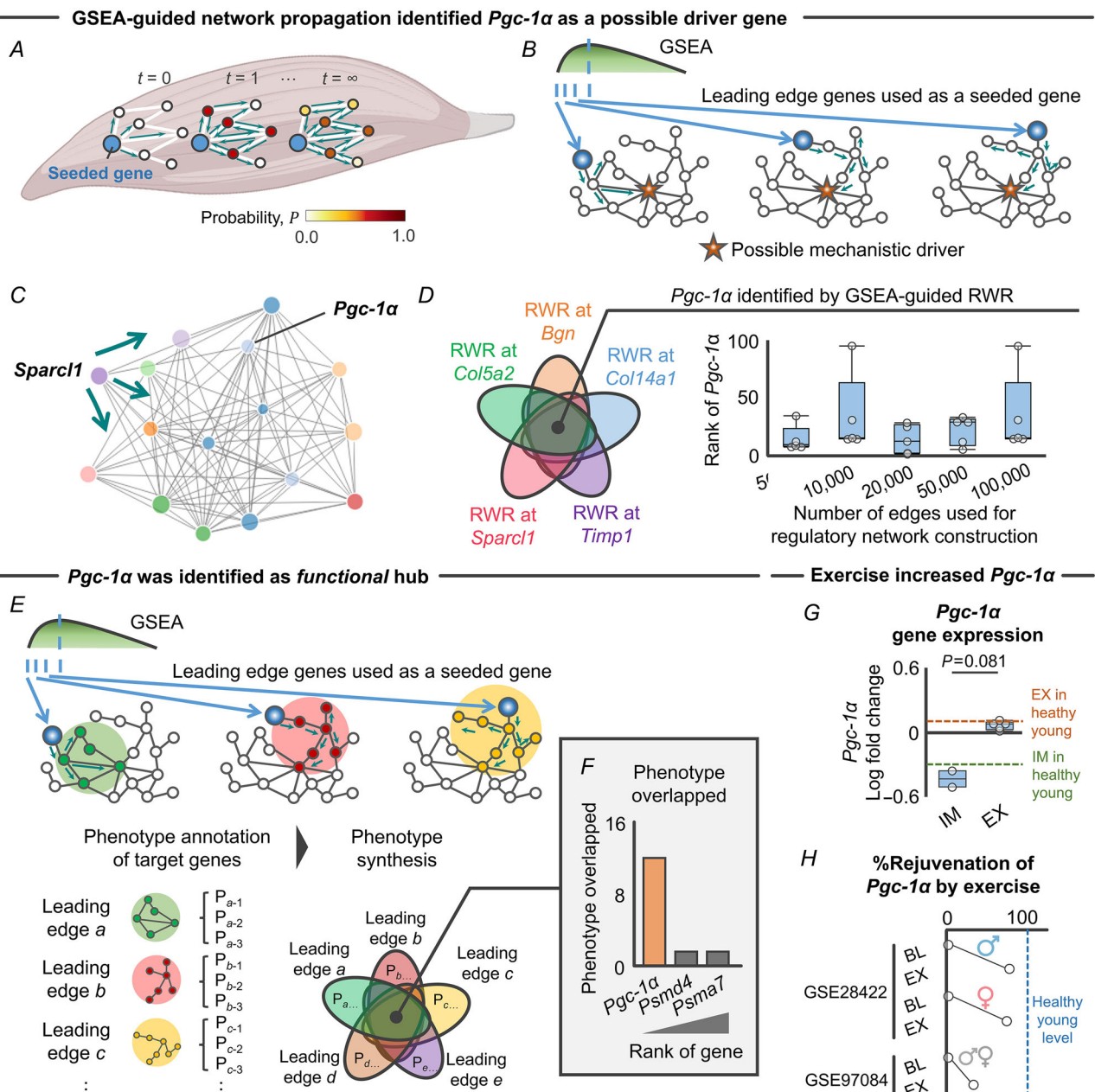

**Figure 6. GSEA-guided network propagation identified *Pgc-1α* as a possible driver gene**

*A*, schematic showing the concept of network propagation on the *de novo* responsive gene regulatory network constructed by WGCNA. *B*, schematic showing the network paradigm of GSEA-guided network propagation starting at the five leading edge genes identified by GSEA. GSEA-guided network propagation implemented multiple RWR with the goal of identifying possible mechanistic drivers (highlighted as red star) of phenotype associated with the leading edge genes. *C*, example of RWR started at *Sparcl1*, illustrating 15 genes in the vicinity of *Sparcl1* that include *Pgc-1α*. *D*, GSEA-guided RWR consistently visited *Pgc-1α* within the top 100 rank of the probability score. *E*, schematic showing GSEA-guided network propagation with subsequent phenotypic (i.e. Wiki pathways) annotation to identify phenotypes associated with the five leading edge genes. *F*, *Pgc-1α* was identified as a functional hub. The graph shows the number of phenotypes overlapping in each gene. *G*, the *Pgc-1α* response was different between the immobilization (*n* = 2) and exercise protocols (*n* = 5). Healthy young values are also provided for visualization purposes (dotted line; exercise: *n* = 3, immobilization: *n* = 2). *H*, %Rejuvenation of *Pgc-1α* by resistance exercise (0%: elderly at baseline, 100%: healthy young adults). Statistical analysis was performed using a Mann–Whitney *U* test (*G*). Abbreviations: BL, baseline; EX, exercise; FAPs, fibro-adipogenic progenitors; GSEA, gene set enrichment analysis; IM, immobilization; RWR, random walk with restart; WGCNA, weighted gene correlation network analysis. [Colour figure can be viewed at wileyonlinelibrary.com]

network construction (i.e. different network topology) may influence the ability of network propagation to identify the *Pgc-1α* node, we performed sensitivity analyses in which a greater number of co-regulated genes pairs (10,000, 20,000, 50,000 and 100,000 edges) were used for the network construction. Regardless of the varied network topologies, *in silico* pseudo-activation of the five genes consistently visited the *Pgc-1α* gene node (Fig. 6*D*). This finding further supports the conclusion that the five leading edge genes are co-regulated with the *Pgc-1α* gene following immobilization and resistance exercise.

To further determine whether *Pgc-1α* serves as 'functional' hub of the FAP adipogenesis gene module, we repeated the RWR starting at the five leading edge genes and determined phenotypes associated with pseudo-activated genes after each network propagation [i.e. significantly enriched Wiki pathways (Slenter et al., 2018); Fig. 6*E*]. The *in silico* overexpression of the five leading edge FAP adipogenesis genes consistently regulated 19 different Wiki pathways, in which *Pgc-1α* was the hub gene contributing to 12 (63.2%) Wiki pathways (Fig. 6*F*). Together, GSEA-guided network propagation identified *Pgc-1α* as a functional hub of the responsive regulatory network for FAP adipogenesis.

Since *Pgc-1α* is upregulated by resistance exercise in elderly muscle to an extent similar to the healthy young counterparts (Fig. 6*G*), we also investigated whether upregulated *Pgc-1α* gene expression in elderly individuals following resistance exercise approximates the level of healthy young adults. For this, we revisited the MetaMEx database and accessed transcriptomic data of skeletal muscle in healthy young individuals without exercise intervention (two studies with three data sets; mean age: 25.1 years old; body mass index: 24.6 kg/m$^2$; $n = 29$ participants) (Raue et al., 2012; Robinson et al., 2017). To account for batch effects induced by intertrial methodological heterogeneity, we used a transcriptomic data set of healthy young participants that was published from the same papers of resistance exercise as the elderly participants (GSE28422 and GSE97084) (Raue et al., 2012; Robinson et al., 2017). We then calculated the '% rejuvenation' of *Pgc-1α* expression (0%: elderly skeletal muscle without exercise, 100%: healthy young muscle without exercise) induced by resistance exercise. The results showed that *Pgc-1α* expression recovered to ∼60% the level of healthy young muscle, although we found intertrial heterogeneity in the *Pgc-1α* response (Fig. 6*H*). Of note, consistent with FAP adipogenesis genes (Fig. 3*C*), male and female participants displayed similar transcriptomic responses of *Pgc-1α* to resistance exercise (Fig. 6*H*). We interpret these data to suggest that, independent of sex, resistance exercise exerts rejuvenating effects on expression of the functional hub gene *Pgc-1α*, ultimately driving downstream signals that decrease IMAT accumulation.

To address the possibility that the findings from these studies are a unique response of mouse cells in which miR-206 has been manipulated, we evaluated FAP signature genes in human elderly skeletal muscle. For this, we accessed single-nuclei RNA-seq data from fatty infiltrated skeletal muscle in human elderly participants (GSE200487) (Fitzgerald et al., 2023). This previous study identified an membrane metallo-endopeptidase (MME)$^+$ FAP subpopulation with high adipogenic potential. Therefore, we defined the 9 genes (*Aldh1a2*, *Col15a1*, *Col4a1*, *Fmo2*, *Itga8*, *Itih5*, *Mme*, *Scn7a*, *Smoc2*) uniquely expressed in this subpopulation as 'MME$^+$ FAP signature genes'. We then repeated ssGSEA, which revealed that resistance exercise and immobilization exert opposing effects on MME$^+$ FAP signature genes (Fig. 7*A*). We also observed a significant linear relationship between the transcriptomic response of human MME$^+$ FAP signature genes and mouse FAP adipogenesis genes under manipulation of miR-206 (Fig. 7*B*). Subsequent leading edge analysis of GSEA revealed that two genes (*Smoc2* and *Col4a1*) were differentially regulated across the exercise and immobilization protocols (Fig. 7*C*). Network propagation starting at *Smoc2* also identified *Pgc-1α* as a hub gene of the network (Fig. 7*D* and *E*). Notably, like *Sparcl1* (identified in the original analysis performed in mice), *Smoc2* is a member of the SPARC family. These results support the conclusion that *Pgc-1α* is a possible master regulator of a human FAP subpopulation with high adipogenic potential.

### *In silico* and *in situ* activation of *Pgc-1α* targets mitochondrial fatty acid oxidation

Taken in the light of previous literature, the findings shown above suggest that *Pgc-1α* is an activity-dependent driver gene of FAP adipogenesis in skeletal muscle. To address the possible downstream signalling of *Pgc-1α*, we again performed RWR to induce *in silico* pseudo-activation of the *Pgc-1α* gene and assessed signalling pathways associated with genes in the vicinity of *Pgc-1α* in the reactive gene regulatory network (Fig. 8*A*). RWR starting at *Pgc-1α* visited 715 genes, which significantly activated 53 different pathways, including those associated with mitochondria-mediated lipid metabolism such as 'Fatty acid beta-oxidation' and 'Mitochondrial LC-fatty acid beta-oxidation' (Fig. 8*B*).

Since the network propagation of *Pgc-1α* activates all pathways in the vicinity of *Pgc-1α*, these pathways do not necessarily mediate the *Pgc-1α*-driven FAP adipogenesis. To address this issue, we integrated the two independent network propagations starting at the exercise-responsive five leading edge FAP adipogenesis genes and *Pgc-1α* (Fig. 8*C*). The assumption of this approach is that overlapping pathways of GSEA-guided (forward) network

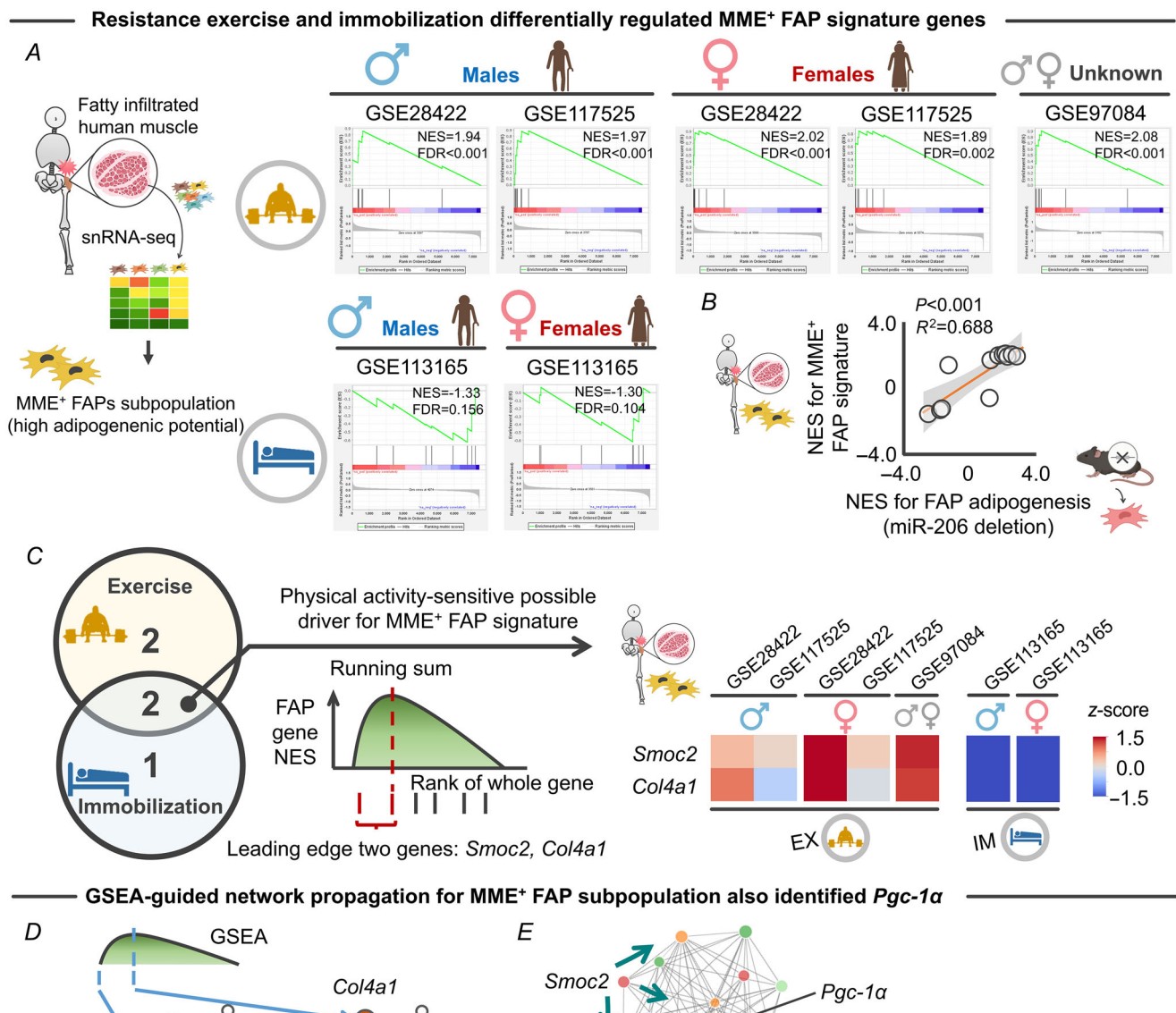

**Figure 7. Transcriptomic response of human MME⁺ FAP signature genes to resistance exercise and immobilization protocols in elderly individuals**

*A*, schematic showing the process of identification of the MME⁺ FAP subpopulation in elderly participants that exhibit high adipogenic potential (GSE200487) (Fitzgerald et al., 2023). The snRNA-seq identified 9 signature genes for the MME⁺ FAP subpopulation (*Aldh1a2*, *Col15a1*, *Col4a1*, *Fmo2*, *Itga8*, *Itih5*, *Mme*, *Scn7a*, *Smoc2*). Single-sample GSEA revealed that resistance exercise and immobilization regulated MME⁺ FAP signature genes in opposing directions, regardless of sex. *B*, transcriptomic response, represented by NES, of human MME⁺ FAP signature genes to immobilization (*n* = 2) and resistance exercise protocols (*n* = 5) was significantly associated with those of mice FAP adipogenesis genes under manipulation of miR-206. NES calculated by GSEA in each condition (i.e. immobilization and exercise) for elderly and young participants was used for the analysis. *C*, leading edge analysis identified two genes (*Smoc2* and *Col4a1*) that overlapped across the exercise and immobilization protocols. Numbers in the Venn diagram represent the number of leading edge genes. Colour in the heatmap indicates the *z*-score for each gene. *D*, schematic showing the network paradigm of GSEA-guided network propagation starting at the two leading edge genes identified by GSEA (*Smoc2* and *Col4a1*). *E*, example of RWR started at *Smoc2*, illustrating 15 genes in the vicinity of *Smoc2* that include *Pgc-1α*. Portions of the figures were created with biorender.com. Statistical analyses were performed using linear regression (*C*). Abbreviations: EX, exercise; FAPs, fibro-adipogenic progenitors; FDR, false discovery rate; GSEA, gene set enrichment analysis; IM, immobilization; NES, normalized enrichment score; snRNA-seq, single-nuclei RNA-seq. [Colour figure can be viewed at wileyonlinelibrary.com]

—— **In silico** activation of **Pgc-1α** promoted mitochondrial fatty acid oxidation ——

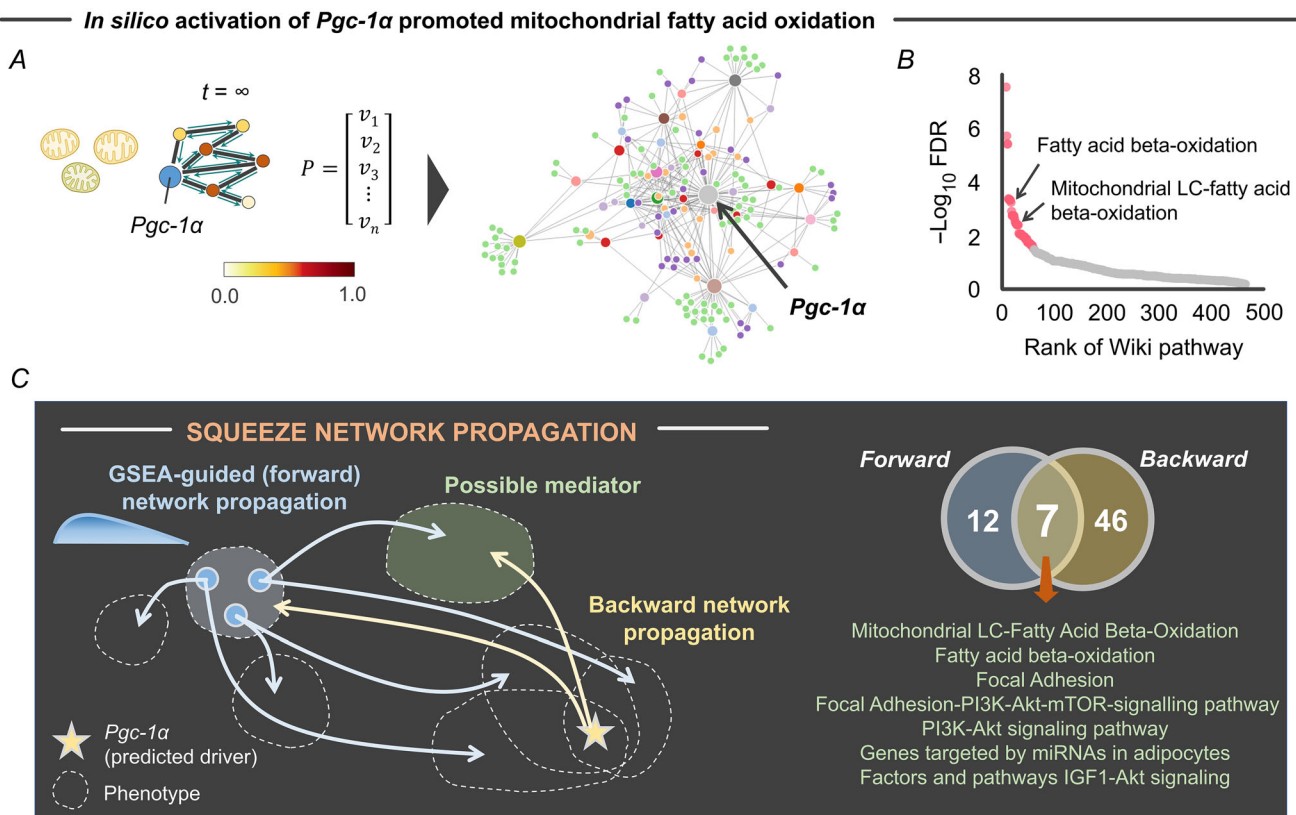

—— Overexpression of **Pgc-1α** **in situ** promoted mitochondrial fatty acid oxidation ——

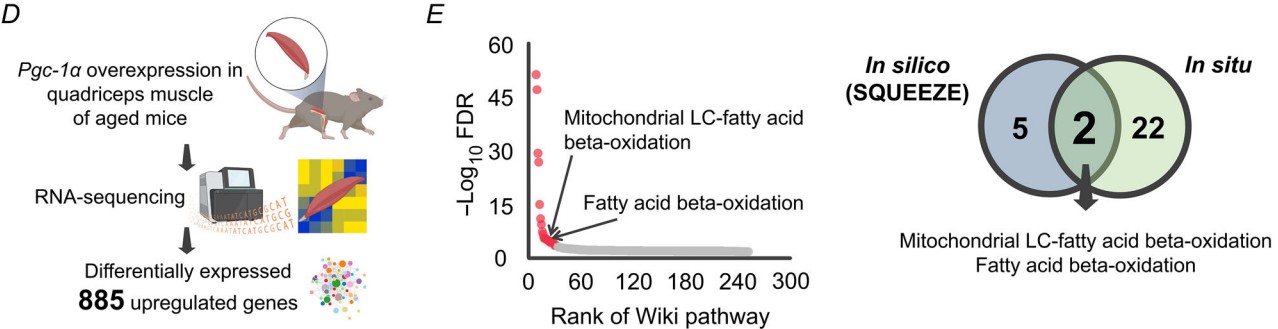

**Figure 8. *In silico* and *in situ* activation of *Pgc-1α* targeted mitochondrial fatty acid oxidation**
*A*, schematic showing RWR started at the *Pgc-1α* gene node (highlighted by blue colour) on the *de novo* gene regulatory network constructed by WGCNA. Gene regulatory network constructed by *Pgc-1α* pseudo-activation is provided. *B*, Wiki pathway enrichment analysis of the gene regulatory network constructed by *Pgc-1α* pseudo-activation. Pathways significantly (FDR < 0.05) modulated are illustrated in red. *C*, schematic illustrating the concept of squeeze network propagation, a bidirectional approach integrating GSEA-guided leading edge (forward) and *Pgc-1α* (backward) network propagations. The squeeze network propagation identified seven Wiki pathways which are possible mediators of downstream signal of *Pgc-1α* towards regulation of FAP adipogenesis. *D*, schematic showing the experimental flow of RNA-seq after *Pgc-1α* overexpression in the quadriceps muscle of aged mice *in situ*, identifying 885 genes significantly upregulated compared to wild-type control aged mice (Garcia et al., 2018). *E*, Wiki pathway enrichment analysis of the *Pgc-1α* target genes. Pathways significantly (FDR < 0.05) modulated are illustrated in red. Two pathways, 'Fatty acid beta-oxidation' and 'Mitochondrial LC-fatty acid beta-oxidation', were consistently modulated by *in silico* and *in situ* activation of *Pgc-1α*. Abbreviations: FAPs, fibro-adipogenic progenitors; FDR, false discovery rate; GSEA, gene set enrichment analysis; RWR, random walk with restart; WGCNA, weighted gene correlation network analysis. [Colour figure can be viewed at wileyonlinelibrary.com]

propagation (i.e. starting at FAP adipogenesis genes) and backward (i.e. starting at *Pgc-1α*) network propagation are possible mediators. This bidirectional network propagation, or 'squeeze network propagation', identified seven pathways, including 'Fatty acid beta-oxidation' and 'Mitochondrial LC-fatty acid beta-oxidation' (Fig. 8*C*).

To further validate the *in silico* network inference, we accessed archived RNA-seq data from aged mice with *Pgc-1α* overexpression in quadriceps muscle, which yielded 885 significantly upregulated genes when compared to wild-type aged mice (Garcia et al., 2018) (Fig. 8*D*). Consistent with *in silico* prediction (Fig. 8*B*), we identified 'Fatty acid beta-oxidation' and 'Mitochondrial LC-fatty acid beta-oxidation' as significant pathways associated with the 885 genes upregulated after *Pgc-1α* activation *in situ* (Fig. 8*E*). Intriguingly, these two pathways were the only overlapping pathways identified by both *in silico* and *in situ* activation of *Pgc-1α* (Fig. 8*E*). These data suggest that these two pathways are mediators of *Pgc-1α*-driven regulation of FAP adipogenesis.

Finally, we investigated whether and how resistance exercise and immobilization regulate mitochondrial fatty oxidation. PCA revealed a clear segregation of the transcriptomic response of genes related to mitochondrial fatty acid oxidation when comparing immobilization and resistance exercise protocols (Fig. 9*A*). Specifically, we found that these genes were upregulated by resistance exercise but downregulated by immobilization (Fig. 9*B*). In these analyses, elderly and healthy young muscle displayed a similar transcriptomic response (Fig. 9*A* and *B*). The responses of these genes to resistance exercise and immobilization protocols were correlated with those of FAP adipogenesis in elderly individuals (Fig. 9*C*). Individual gene-level analysis revealed that physical activity level was positively associated with expression of key genes involved in mitochondrial fatty acid oxidation (Fig. 9*D*). These analyses support the hypothesis that resistance exercise upregulates *Pgc-1α* and promotes mitochondrial fatty acid oxidation to suppress FAP adipogenesis, but that immobility has the opposite effect.

## Discussion

Physical exercise has been a widely recognized pillar in the management of skeletal muscle health in older individuals. Accumulating clinical evidence has shown that physical exercise reduces ectopic deposition of fat in skeletal muscle in the elderly and in patients with chronic diseases (Tuñón-Suárez et al., 2021). However, the molecular mechanisms underlying the adipose-regulating effects of exercise have been poorly understood, thereby representing a critical knowledge gap in the development

of effective health-promoting rehabilitative strategies for this population. Here, we took an unbiased approach with a series of computational analyses to explore a shared mechanism of exercise-induced IMAT regulation (see Abstract figure). After identifying a set of genes associated with FAP adipogenesis from publicly archived data sets, we discovered that resistance exercise and immobilization in aged skeletal muscle displayed opposing regulatory effects on FAP adipogenesis. Integrated analysis of WGCNA and GSEA-guided network propagation revealed that the exercise- and immobilization-driven regulation on FAP adipogenesis was co-expressed with the *Pgc-1α* gene response, where *Pgc-1α* acted as a functional hub for the larger responsive gene regulatory network. *In silico* network inference of squeeze network propagation integrated with *in situ* activation of *Pgc-1α* identified mitochondrial fatty acid oxidation as a mediator of *Pgc-1α*-driven regulation of FAP adipogenesis. These findings suggest that physical activity reduces IMAT deposition via upregulation of *Pgc-1α*-mediated mitochondrial fatty acid oxidation and subsequent inhibition of FAP adipogenesis.

The relationship between FAP adipogenesis, a primary inducer of IMAT (Heredia et al., 2013; Uezumi et al., 2010, 2011), and chronic physical exercise, a known suppressor of IMAT (Tuñón-Suárez et al., 2021), has been poorly understood. To address this critical shortcoming, this study identified genes associated with FAP adipogenesis and performed GSEA to demonstrate that chronic resistance exercise and immobilization differentially regulated FAP adipogenesis. This trend was consistent across the different transcriptomic data sets of resistance exercise in elderly individuals, indicating that the relationship between exercise-responsive genes and FAP adipogenesis is robust. Since the adipogenic inhibitor, *Sparcl1*, was identified as the primary leading edge gene upregulated by resistance exercise, we interpreted this finding to suggest that upregulation of *Sparcl1* in skeletal muscle contributes to exercise-induced IMAT inhibition. Interestingly, the regulatory effects of FAP adipogenesis by resistance exercise and immobilization were comparable across the male and female participants. Our results are in line with previous meta-analyses demonstrating that both males and females display a significant reduction in visceral fat after exercise (Vissers et al., 2013). Nevertheless, there are conflicting results on the effects of sex on the mobilization and oxidation of endogenous triglycerides during exercise (Arner et al., 1990; Burguera et al., 2000; Hellström et al., 1996). Sex-dependent regulation of lipid metabolism by exercise remains an interesting area for future investigation.

To interrogate potential mechanisms underlying the effects of exercise and immobilization on FAP lineage specification, this study introduced a network paradigm, GSEA-guided network propagation, under the

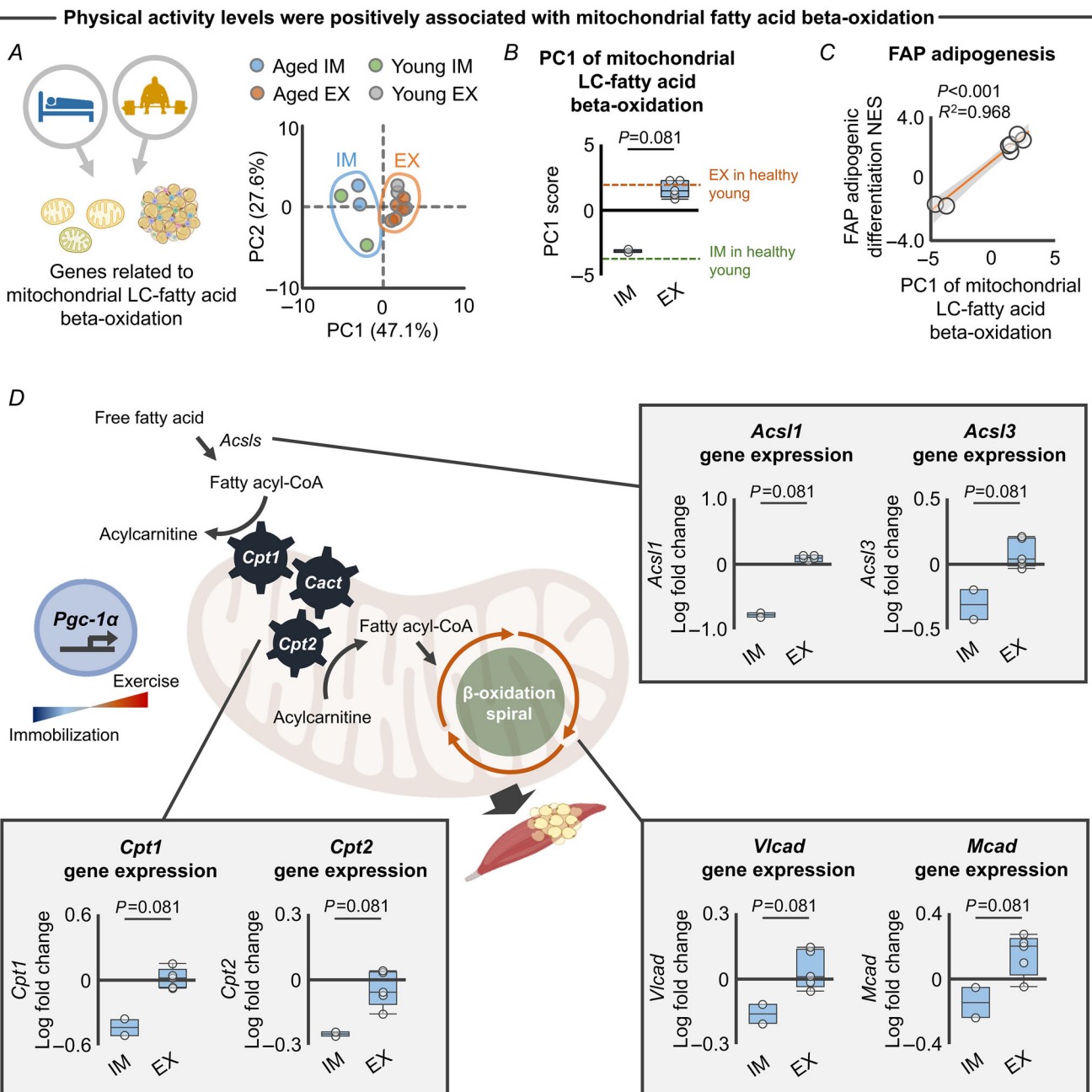

**Figure 9. Individual gene responses of mitochondrial LC-fatty acid beta-oxidation to exercise and immobilization protocols**

*A*, PCA showing the separate clusters in immobilization (blue area; young: *n* = 2, aged: *n* = 2) and resistance exercise (orange area; young: *n* = 3, aged: *n* = 5) against overall expression of genes related to mitochondrial LC-fatty acid beta-oxidation. *B*, PC1 extracted from the PCA revealed a distinct transcriptomic response of the genes related to mitochondrial LC-fatty acid beta-oxidation to the immobilization (*n* = 2) and resistance exercise protocols (*n* = 5) in elderly individuals. Healthy young values are also provided for visualization purposes (dotted line; exercise: *n* = 3, immobilization: *n* = 2). *C*, transcriptomic response of the genes related to mitochondrial LC-fatty acid beta-oxidation to the immobilization (*n* = 2) and resistance exercise protocols (*n* = 5) was significantly associated with regulation of FAP adipogenesis (i.e. NES). NES is calculated by GSEA (see Fig. 3*C*). *D*, transcriptomic response of key individual genes involved in mitochondrial LC-fatty acid beta-oxidation to the immobilization (*n* = 2) and resistance exercise protocols (*n* = 5). Statistical analyses were performed using a Mann–Whitney *U* test (*B*, *D*) and linear regression (*C*). Abbreviations: EX, exercise; FAPs, fibro-adipogenic progenitors; GSEA, gene set enrichment analysis; IM, immobilization; NES, normalized enrichment score; PC, principal component; PCA, principal component analysis. [Colour figure can be viewed at wileyonlinelibrary.com]

assumption of a 'disease module hypothesis' (Barabási et al., 2011), where disease-associated genes or proteins probably share the same topological neighbourhood in a network (Barabási et al., 2011). In this study, we used a GSEA leading edge as a start of network propagation on the WGCNA-based responsive gene regulatory network to explore upstream regulators of exercise-induced FAP adipogenesis. These analyses successfully identified the transcription cofactor gene, *Pgc-1α*, as differentially regulated by immobilization and exercise. *Pgc-1α* has been known as a master regulator of mitochondrial biogenesis in the skeletal muscle response to exercise (Handschin & Spiegelman, 2008). Building upon these previous findings, the present study suggests a previously unappreciated mechanistic role of *Pgc-1α* in the regulation of FAP adipogenesis and IMAT accumulation in aged skeletal muscle. These findings support the utility of GSEA-guided network propagation as a powerful tool to predict the transcriptional regulator of skeletal muscle health. This network propagation-based approach is further strengthened by combining another type of network propagation that we first introduced as 'squeeze network propagation'. This bidirectional network propagation allows for discovery of novel candidate mediators between the seeded two genes or phenotypes. Through this analysis, we successfully prioritized possible pathways and identified mitochondrial fatty oxidation as a primary driver for FAP regulation by *Pgc-1α*.

When compared to healthy young adults, the transcriptomic response of elderly individuals displayed higher enrichment to FAP adipogenesis. On the other hand, the transcriptomic response of *Pgc-1α* and downstream mitochondrial fatty acid beta-oxidation to physical activity was comparable across young and elderly individuals. This discordance implies different regulatory mechanisms of FAP adipogenesis across different ages. FAPs sense and integrate niche signals that keep their differentiation potential (Giuliani et al., 2022). Indeed, the disruption of these niche cues, such as persistent age-related low-grade inflammation, leads to FAP differentiation into adipocytes and fibroblasts (Heredia et al., 2013; Uezumi et al., 2010, 2011). Nevertheless, the detailed mechanism regulating the adipogenic potential of FAPs by physical activity remains underexplored. Our findings highlight the need for more mechanistic studies aimed at elucidating the age-dependent mechanism by which physical activity levels regulate FAP fate in skeletal muscle.

It is noteworthy that FAP adipogenesis induced by miR-206 deletion was accompanied by extracellular matrix remodelling, as documented by significant changes in focal adhesion signalling. Accordingly, the five leading edge genes that were predominantly regulated by exercise and immobilization code for extracellular matrix-related proteins. As such, it is possible that matrix remodelling

mediates the effects of exercise on the regulation of the FAP adipogenesis programme. Indeed, there is evidence that FAP differentiation is regulated by biophysical properties of extracellular matrix (Loomis et al., 2022). Dysregulation of FAP differentiation also leads to fibrosis, a leading cause of matrix stiffening (Loomis & Smith, 2023). This bidirectional interaction between FAPs and their surrounding matrix, a phenomenon known as dynamic reciprocity (Bissell et al., 1982), is an area of interest for future studies.

Although this study provides novel insight into the pathogenesis of age-related IMAT accumulation, it has limitations. First, the findings of this study are exploratory in nature and are based on a small number of currently available data sets that employ an immobilization model. This may contribute to bias depending on the participants' characteristics and the methodology used in the original studies. Previous studies have indicated an association between obesity [i.e. high body mass index (BMI)] and IMAT (Goodpaster et al., 2000; Malenfant et al., 2001). As such, we cannot discount the possibility that the impact of physical activity level on FAP adipogenesis may be a function of participants' body mass. Nevertheless, the averaged BMI for participants across resistance exercise and immobilization protocols was similar, indicating a probable minimal impact of BMI on the distinct FAP regulatory effects observed across the two protocols. Another limitation is that the possible working mechanism proposed in this study is based on transcript levels only. An important next step will be to interrogate the identified mechanism at the protein and tissue levels. Finally, FAP adipogenesis genes were defined according to archived RNA-seq data from mouse skeletal muscle under manipulation of miR-206 (Wosczyna et al., 2021). Although growing evidence indicates that murine FAPs have a transcriptomic profile similar to human FAPs (Fitzgerald et al., 2023), the proposed mechanisms may not completely capture the dynamics of FAP adipogenesis observed in elderly skeletal muscle. Nevertheless, the current study demonstrated that exercise and immobilization protocols exert opposing effects on signature genes of human FAPs that exhibit high adipogenic potential, providing novel mechanistic insight into how different physical activity levels regulate IMAT in elderly skeletal muscle.

While this study focused on skeletal muscle, a major conceptual innovation of the proposed method is the demonstration of the mechanistic link between exercise-driven health-promoting effects and target disease (i.e. IMAT in elderly individuals). It is widely recognized that physical exercise elicits systemic benefits beyond skeletal muscle in both young and older populations (Thyfault & Bergouignan, 2020). Yet, the mechanisms of the multisystem benefits induced by exercise remain largely unknown. We anticipate that

the network paradigm proposed in this study may have broader implications in the field of exercise physiology and ageing research.

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

## Additional information

### Data availability statement

The raw data that support the experimental findings are included in the supplementary material. Any additional information required to reanalyse the data reported in this paper is available from the corresponding author upon request.

### Competing interests

The authors declare no competing interests.

### Author contributions

All authors made substantial contributions in the following areas: (1) conception and design of the study, acquisition of data, analysis and interpretation of data, drafting of the article; (2) final approval of the article version to be submitted; and (3) agreement to be personally accountable for the author's own contributions and to ensure that questions related to the accuracy are appropriately investigated, resolved and the resolution documented in the literature. The specific contributions of the authors are as follows: H.I., F.A. and Y.M. provided the concept, idea and experimental design for the studies. H.I., F.A. and Y.M. wrote the manuscript. H.I., F.A. and Y.M. provided data collection, analyses, interpretation and review of the manuscript. H.I. obtained funding for the studies.

### Funding

This study was supported in part by (1) a JSPS KAKENHI (Grant number: 23H03308) from the Japan Society for the Promotion of Science (https://www.jsps.go.jp/) to H.I. and (2) Tokai Pathways to Global Excellence Project (T-GEx) (https://www.t-gex.nagoya-u.ac.jp/en/) to H.I. The funders had no role

in study design, data collection and analysis, decision to publish or preparation of the manuscript.

## Keywords

ageing, exercise, fatty acid, gene regulatory networks, immobilization, mitochondria, skeletal muscle, systems biology

## Supporting information

Additional supporting information can be found online in the Supporting Information section at the end of the HTML view of the article. Supporting information files available:

**Peer Review History**
**Raw data**

