## [Peer Review History · The Journal of Physiology]

Network-based systematic dissection of exercise-induced inhibition of myosteatorsis in older individuals

Hirofaka Iijima, Fabrisia Ambrosio, and Yusuke Matsui
DOI: 10.1113/JP285349

Corresponding author(s): Hirofaka Iijima (hijima1@mgh.harvard.edu)

The following individual(s) involved in review of this submission have agreed to reveal their identity: Ian R Lanza (Referee #1); Allison Owen (Referee #2); Sean Williams (Referee #3)

Review Timeline:

Submission Date:	22-Jul-2023
Editorial Decision:	29-Aug-2023
Revision Received:	27-Sep-2023
Editorial Decision:	23-Oct-2023
Revision Received:	07-Nov-2023
Accepted:	10-Nov-2023

Senior Editor: Paul Greenhaff

Reviewing Editor: Christopher Sundberg

Transaction Report:

Dear Dr Iijima,

Re: JP-RP-2023-285349 "Network-based systematic dissection of exercise-induced inhibition of myosteatosis in older individuals" by Hiroataka Iijima, Fabrisia Ambrosio, and Yusuke Matsui

Thank you for submitting your manuscript to The Journal of Physiology. It has been assessed by a Reviewing Editor and by 2 expert referees and we are pleased to tell you that it is potentially acceptable for publication following satisfactory major revision.

REVISION CHECKLIST:

We look forward to receiving your revised submission.

Yours sincerely,

Paul Greenhaff
Senior Editor
The Journal of Physiology

REQUIRED ITEMS

-Author photo and profile. First (or joint first) authors are asked to provide a short biography (no more than 100 words for one author or 150 words in total for joint first authors) and a portrait photograph. These should be uploaded and clearly labelled with the revised version of the manuscript. See Information for Authors for further details.

-The Journal of Physiology funds authors of provisionally accepted papers to use the premium BioRender site to create high resolution schematic figures. Follow this link and enter your details and the manuscript number to create and download figures. Upload these as the figure files for your revised submission. If you choose not to take up this offer we require figures to be of similar quality and resolution. If you are opting out of this service to authors, state this in the Comments section on the Detailed Information page of the submission form. The link provided should only be used for the purposes of this submission. Authors will be charged for figures created on this premium BioRender account if they are not related to this manuscript submission.

-Please upload separate high-quality figure files via the submission form.

EDITOR COMMENTS

Reviewing Editor:

Thank you for submitting your manuscript to The Journal of Physiology to be considered for inclusion in the special issue on the physiology of ageing skeletal muscle and the protective effects of exercise. Two expert reviewers have assessed your manuscript and while both were complimentary on the quality of writing and the potential impact of the study, they both identified several major concerns. Specifically, centering the analyses around the 161 differentially expressed mouse genes in response to a single micro-RNA (miR-206) manipulation may not translate well to humans and/or may bias the data away from important genes that were not captured by manipulation of miR-206. Careful consideration should also be given to the reporting of the bioinformatics and statistical approaches used in the study to help ensure repeatability of the analyses.

Senior Editor:

This manuscript has been considered by two expert reviewers and a reviewing editor. All see merits in the work, but some major concerns have been raised which the authors are invited to address. The most important concerns that must be fully addressed involves the approach of centring analyses around a single study that revealed 161 differentially expressed mouse genes in response to a single micro-RNA (miR-206) manipulation. This on the face of it does not seem to be an overly robust approach as it may not translate well to the human condition given the mouse is not considered to be a robust model of human ageing (doi: 10.1038/sj.embor.7400422) and/or may bias the data away from important genes that were not captured by manipulation of miR-206. It is also important that any revised version should give careful consideration to the

reporting of the bioinformatics and statistical approaches used in the study to help ensure repeatability of the analyses. Hopefully the authors feel they can fully address these major concerns.

REFeree COMMENTS

Referee #1:

This is a very well written manuscript that focusses on the important and understudied topic of myosteatosis and the mechanisms by which intermuscular adipose tissue (IMAT) is influenced by aging and exercise. This work has the potential to be impactful because IMAT accumulation is a feature of aging while exercise has been shown to reduce IMAT and associated derangements in muscle and metabolic health. The main contribution of this work is in unraveling the molecular/cellular pathways by which aging and exercise influence IMAT. Toward this goal, the authors thoughtfully leveraged publicly available gene expression datasets implemented a number of bioinformatic tools to explore potential regulatory networks involved in the regulation of fibro-adipogenic progenitor cells (FAPs). The manuscript is interesting and addresses a timely and impactful topic. The manuscript is a nice example of the value of publicly available datasets. The authors are commended on crafting this manuscript in a way that is clear and understandable to readers without high-level knowledge of some of the sophisticated bioinformatics approaches that they used. Notwithstanding, there are several aspects of this work that need to be carefully considered:

1. The manuscript centers around the concept that FAPs differentiate into adipocytes within muscle tissue, giving rise to IMAT. This concept has growing support, but no studies have identified the transcriptional drivers of FAPs-to-IMAT. The authors build a story that is based largely on one study where a specific micro RNA (miR-206) was shown to influence adipogenic differentiation of FAPs. Based on that single earlier study in mice, they identified 161 genes that were differentially expressed in FAPs in the presence/absence of siRNA to miR-206. The authors claim that these genes drive FAP differentiation to adipocytes; an assumption that forms the foundation of this manuscript. While the earlier paper convincingly links miR-206 to FAP differentiation to adipocytes through Runx1, the assumption that those 161 genes are also involved in FAP differentiation to adipocytes may or may not be true. The risk in hanging ones' hat on a single experiment for this type of analysis is that they may be completely missing the important regulators in humans vs mice or even errantly focusing on promiscuous molecules that are associated but not directly linked.
2. Somewhat related to the previous comment, did the authors observe a difference in miR-206 levels in young vs. older adults or trained vs. untrained participants in the METAMEX data that could explain the significantly elevated FAPs NES score figure 2D?
3. The authors pick out Sparcl1 as one of the 5 leading edge genes that were significantly changed across exercise and immobilization protocols. The potential importance of Sparcl1 in IMAT formation has been shown already, making it a logical molecule of interest, but as the authors point out, Sparcl1 is also involved in many physiological processes other than adipogenesis. At most, the identification of Sparcl1 in this study is confirmatory.
4. Using the other 4 leading edge genes, the authors then take a data-driven approach to construct a regulatory gene network that they claim to play a role in FAPs conversion to adipocytes. While this is not an unreasonable approach to develop new hypotheses and candidate pathways/networks linked with those 4 molecules, the risk is in inappropriately ascribing uncovered networks to a specific physiological process (i.e, adipogenesis) when those 4 genes are perhaps just red herrings. Nevertheless, pgc1a was identified as a hub gene and proposed to be a central regulator of FAPs adipogenesis. Despite extensive in silico and in situ modeling to confirm/validate that the 5 leading edge genes co-regulate with the pgc1a node with exercise and immobilization, it is difficult to ignore the possibility that 4-5 genes identified from the 161 differentially expressed mouse genes in response to a single miRNA manipulation may not translate well into the human situation.

Referee #2:

This manuscript by Iijima et al aims to leverage existing transcriptomic data sets to unveil mechanistic drivers of FAP adipogenesis in the aged which are reversed, at least in part, by exercise intervention. The authors conducted analysis on several data sets of human studies as well as murine investigations which together elucidated PGC1 α as a critical hub which is upregulated by exercise and may downregulate adipogenesis in the elderly. Using in silico analysis, they go on to show that PGC1 α -mediated regulation of mitochondrial fatty acid oxidation may be a key inhibitor of FAP adipogenesis and thus fat accumulation in muscle. The authors should be complimented for sophisticated analysis using existing data sets and well-written manuscript to contribute a novel finding regarding the important topic regarding skeletal muscle health in the aged. However, some concerns reduce enthusiasm for the work put forth, as detailed in a point-by-point fashion below.

Major

1. Although the authors describe additional filtering strategies for low read counts (which is not defined) and signal intensity, there is some reservation whether different read depths were corrected for to make the data comparable among the different studies and datasets. It would be helpful to have a bioinformatic statistician review the conducted analyses given their complex nature.
2. It is appreciated that the authors communicated a clear story regarding exercise-mediated regulation of insulin sensitivity and FAP adipogenesis. However, in that focus there seems to be a missed opportunity to communicate other potentially interesting findings which would be of use to the field. In Figure 1, the authors immediately filtered for insulin sensitivity-related genes prior to conducting PCA analysis. Could the authors conduct exploratory PCA and enrichment analyses on the most up- and down-regulated DEGs without filtering for insulin-sensitivity genes? Not only would it be interesting to determine if the grouping relationships on the PCA are still evident even when not selecting for specific genes, but also would be interesting to see if the most differentially regulated enrichment pathways are metabolism-related, thus giving further support to the current study.
3. Similar to point 2, could the authors provide the other differentially regulated pathways identified in Fig. 5 B and E as supplemental information? Although a straightforward story is appreciated, it seems like there is data that could be advantageous to readers that is currently not communicated in its current form.
4. A fair degree of weight is placed upon the FAP adipogenesis-related dataset provided by Wosczyzna et al 2021 in which IMAT is regulated by miR-2016 (mimicry or inhibition) which the authors conducted analysis on to identify a list of 161 genes that "drive adipocytic differentiation of FAPs." This potentially problematic since these DEGs are only those regulated under manipulation of miR-2016 (to my understanding), and thus the list is likely biased and limited since other genes surely may play a role in FAP adipogenesis but were simply not captured by manipulation of miR-2016. This is critical since this DEG list is the foundation for their study. The authors should consider providing the list of 161 DEGs as a supplement, identify genes or conduct enrichment pathway analysis which instill further confidence in the list, and provide discussion of this limitation.
5. Of the 5 leading-edge genes identified (first described in Fig. 2D; Bgn, Col5a2, Col14a1, Sparcl1, and Timp1), it is understood why the authors focused on Sparcl1 given its suggested role in adipogenesis. Collectively, however, this list seems to indicate extracellular matrix remodeling as the general function of these leading-edge genes. This would logically make sense if exercise mediates upregulation of ECM remodeling which then in turn would prevent FAP adipogenic programs. Do the authors think activation of ECM programming diametrically opposes adipogenesis?
6. Could the authors provide additional information on "module 8" in Figure 3 given that one of the 5 identified leading-edge genes (Col5a2) was enriched in this module and may provide further insight to the mechanism by which exercise regulates FAP adipogenesis?

Minor

1. Could the authors indicate study exclusion criteria in the manuscript text to accompany/expand upon rationale for the information provided in Figure 1A?
2. Quite a bit of discussion is added throughout the results section, which is a stylistic choice and is helpful at times, but distracting at others. The authors should consider shortening lines 228-237 and 280-289 which are not necessary to logically follow the respective subsequent results sections.
3. The authors should define WGCNA in line 254.
4. Throughout the manuscript, the color blue notates immobilization and the color yellow notates exercise intervention. Thus, using this same color scheme (blue and yellow) to notate sex (male and female) in Figure 1B is confusing.
5. Figure 3E is confusing in its current form and doesn't seem to add much to the information already provided in Figure 3D. The authors should consider editing or removing.

END OF COMMENTS

This is a very well written manuscript that focusses on the important and understudied topic of myosteatosis and the mechanisms by which intermuscular adipose tissue (IMAT) is influenced by aging and exercise. This work has the potential to be impactful because IMAT accumulation is a feature of aging while exercise has been shown to reduce IMAT and associated derangements in muscle and metabolic health. The main contribution of this work is in unraveling the molecular/cellular pathways by which aging and exercise influence IMAT. Toward this goal, the authors thoughtfully leveraged publicly available gene expression datasets implemented a number of bioinformatic tools to explore potential regulatory networks involved in the regulation of fibro-adipogenic progenitor cells (FAPs). The manuscript is interesting and addresses a timely and impactful topic. The manuscript is a nice example of the value of publicly available datasets. The authors are commended on crafting this manuscript in a way that is clear and understandable to readers without high-level knowledge of some of the sophisticated bioinformatics approaches that they used. Notwithstanding, there are several aspects of this work that need to be carefully considered:

1. The manuscript centers around the concept that FAPs differentiate into adipocytes within muscle tissue, giving rise to IMAT. This concept has growing support, but no studies have identified the transcriptional drivers of FAPs-to-IMAT. The authors build a story that is based largely on one study where a specific micro RNA (miR-206) was shown to influence adipogenic differentiation of FAPs. Based on that single earlier study in mice, they identified 161 genes that were differentially expressed in FAPs in the presence/absence of siRNA to miR-206. The authors claim that these genes drive FAP differentiation to adipocytes; an assumption that forms the foundation of this manuscript. While the earlier paper convincingly links miR-206 to FAP differentiation to adipocytes through Runx1, the assumption that those 161 genes are also involved in FAP differentiation to adipocytes may or may not be true. The risk in hanging ones' hat on a single experiment for this type of analysis is that they may be completely missing the important regulators in humans vs mice or even errantly focusing on promiscuous molecules that are associated but not directly linked.
2. Somewhat related to the previous comment, did the authors observe a difference in miR-206 levels in young vs. older adults or trained vs. untrained participants in the METAMEX data that could explain the significantly elevated FAPs NES score figure 2D?
3. The authors pick out Sparcl1 as one of the 5 leading edge genes that were significantly changed across exercise and immobilization protocols. The potential importance of Sparcl1 in IMAT formation has been shown already, making it a logical molecule of interest, but as the authors point out, Sparcl1 is also involved in many physiological processes other than adipogenesis. At most, the identification of Sparcl1 in this study is confirmatory.
4. Using the other 4 leading edge genes, the authors then take a data-driven approach to construct a regulatory gene network that they claim to play a role in FAPs conversion to adipocytes. While this is not an unreasonable approach to develop new hypotheses and candidate pathways/networks linked with those 4 molecules, the risk is in inappropriately ascribing uncovered networks to a specific physiological process (i.e, adipogenesis) when those 4 genes are perhaps just red herrings. Nevertheless, pgc1a was identified as a hub gene and proposed to be a central regulator of FAPs

adipogenesis. Despite extensive in silico and in situ modeling to confirm/validate that the 5 leading edge genes co-regulate with the *pgc1a* node with exercise and immobilization, it is difficult to ignore the possibility that 4-5 genes identified from the 161 differentially expressed mouse genes in response to a single miRNA manipulation may not translate well into the human situation.

We are very grateful to the reviewers for providing such valuable, thoughtful, and thorough feedback. The comments and questions raised have prompted us to consider our data in new and interesting ways, the culmination of which we feel has greatly strengthened the work. Thank you for your time.

Reviewer #1**[Comment #1]**

This is a very well written manuscript that focusses on the important and understudied topic of myosteatosis and the mechanisms by which intermuscular adipose tissue (IMAT) is influenced by aging and exercise. This work has the potential to be impactful because IMAT accumulation is a feature of aging while exercise has been shown to reduce IMAT and associated derangements in muscle and metabolic health. The main contribution of this work is in unraveling the molecular/cellular pathways by which aging and exercise influence IMAT. Toward this goal, the authors thoughtfully leveraged publicly available gene expression datasets implemented a number of bioinformatic tools to explore potential regulatory networks involved in the regulation of fibro-adipogenic progenitor cells (FAPs). The manuscript is interesting and addresses a timely and impactful topic. The manuscript is a nice example of the value of publicly available datasets. The authors are commended on crafting this manuscript in a way that is clear and understandable to readers without high-level knowledge of some of the sophisticated bioinformatics approaches that they used. Notwithstanding, there are several aspects of this work that need to be carefully considered:

1. The manuscript centers around the concept that FAPs differentiate into adipocytes within muscle tissue, giving rise to IMAT. This concept has growing support, but no studies have identified the transcriptional drivers of FAPs-to-IMAT. The authors build a story that is based largely on one study where a specific micro RNA (miR-206) was shown to influence adipogenic differentiation of FAPs. Based on that single earlier study in mice, they identified 161 genes that were differentially expressed in FAPs in the presence/absence of siRNA to miR-206. The authors claim that these genes drive FAP differentiation to adipocytes; an assumption that forms the foundation of this manuscript. While the earlier paper convincingly links miR-206 to FAP differentiation to adipocytes through Runx1, the assumption that those 161 genes are also involved in FAP differentiation to adipocytes may or may not be true. The risk in hanging ones' hat on a single experiment for this type of analysis is that they may be completely missing the important regulators in humans vs mice or even errantly focusing on promiscuous molecules that are associated but not directly linked.

[Author action #1]

We are very grateful for this critical constructive feedback. We received a similar comment from Reviewer #2 (see **Comment #10**). To support that 161 genes significantly changed after miR-206 manipulation may be involved in FAPs adipogenesis program, we performed Wiki pathway enrichment analysis for the 161 genes. We found that adipogenesis-related pathways, “Adipogenesis” and “PPAR signaling pathway”, were the top 2 significant pathways significantly regulated by miR-206 KO (**Figure 2B**). We have explained these results accordingly (see **Page 8-9 / Line 202-203 in RESULTS**).

Nevertheless, as Reviewer #1 pointed out, we cannot exclude the possibility of bias resulting from the methodology used for miR-206 manipulation in mice. While a recent study showed a transcriptomic similarity across murine and human FAPs¹, there is accumulating evidence for the heterogeneity of human FAPs. Indeed, human FAPs consist of different subpopulations with varying adipogenicity, with MME⁺ FAPs displaying the highest adipogenic potential¹. As such, we sought to determine whether exercise and immobilization protocols differentially regulate the 10 genes unique to the MME⁺ FAPs subpopulation (*Aldh1a2*, *Col15a1*, *Col4a1*, *Fmo2*, *Itga8*, *Itih5*, *Mme*, *Scn7a*, *Smoc2*). We defined these genes as “MME⁺ FAPs signature genes”. We then accessed single-nuclei RNA-seq from fatty infiltrated skeletal muscle in human elderly participants (GSE200487)¹ and repeated single sample GSEA. We found that resistance

exercise and immobilization differentially regulate these MME⁺ FAP signature genes (**Figure S3A**). Further, we demonstrated a significant linear relationship between the transcriptomic response of human MME⁺ FAPs signature genes and mouse FAP adipogenesis genes under manipulation of miR-206 (**Figure S3B**). Subsequent leading-edge analysis of GSEA revealed that two genes (*Smoc2* and *Col4a1*) were differentially regulated across the exercise and immobilization protocols (**Figure S3C**). Intriguingly, network propagation starting at *Smoc2*, also identified *Pgc-1a* as a hub gene of the network (**Figure S3D, E**). Notably, like *Sparcl1* (identified in the original analysis performed in mice), *Smoc2* is a member of the SPARC family. These results support the conclusion that *Pgc-1a* is a possible master regulator of a human FAP subpopulation with high adipogenic potential. We have provided these results as follows:

Page 14-15 / Line 339-355 in RESULTS:

*“To address the possibility that the findings from these studies are a unique response of mouse cells in which miR-206 has been manipulated, we evaluated FAP signature genes in human elderly skeletal muscle. For this, we accessed single-nuclei RNA-seq from fatty infiltrated skeletal muscle in human elderly participants (GSE200487; Fitzgerald et al., 2023). This previous study identified an MME⁺ FAPs subpopulation with high adipogenic potential. Therefore, we defined the 10 genes (*Aldh1a2*, *Col15a1*, *Col4a1*, *Fmo2*, *Irga8*, *Itih5*, *Mme*, *Scn7a*, *Smoc2*) uniquely expressed in this subpopulation as “MME⁺ FAPs signature genes”. We then repeated ssGSEA, which revealed that resistance exercise and immobilization exert opposing effects on MME⁺ FAP signature genes (Figure S3A). We also observed a significant linear relationship between the transcriptomic response of human MME⁺ FAPs signature genes and mouse FAP adipogenesis genes under manipulation of miR-206 (Figure S3B). Subsequent leading-edge analysis of GSEA revealed that two genes (*Smoc2* and *Col4a1*) were differentially regulated across the exercise and immobilization protocols (Figure S3C). Network propagation starting at *Smoc2*, also identified *Pgc-1a* as a hub gene of the network (**Figure S3D, E**). Notably, like *Sparcl1* (identified in the original analysis performed in mice), *Smoc2* is a member of the SPARC family. These results support the conclusion that *Pgc-1a* is a possible master regulator of a human FAP subpopulation with high adipogenic potential.”*

We have also acknowledged the potential discrepancy between murine and human genes as a limitation as follows:

Page 20 / Line 484-492 in DISCUSSION:

“Finally, FAP adipogenesis genes were defined according to archived RNA-seq data from mouse skeletal muscle under manipulation of miR-206 (Wosczyzna et al., 2021). Although growing evidence indicates that murine FAPs have a transcriptomic profile similar to human FAPs (Fitzgerald et al., 2023), the proposed mechanisms may not completely capture the dynamics of FAP adipogenesis observed in elderly skeletal muscle. Nevertheless, the current study demonstrated that exercise and immobilization protocols exert opposing effects on signature genes of human FAPs that exhibit high adipogenic potential, implicating a novel mechanistic insight into how different physical activity levels regulate IMAT in elderly skeletal muscle.”

Figure 2B. Top five Wiki pathways significantly enriched to 161 differentially expressed genes associated with FAP adipogenesis.

Figure S3. Transcriptomic response of human MME⁺ FAP signature genes to resistance exercise and immobilization protocols in elderly individuals

Reference

1. Fitzgerald G, et al. MME(+) fibro-adipogenic progenitors are the dominant adipogenic population during fatty infiltration in human skeletal muscle. *Commun Biol* 2023;6:111.

[Comment #2]

2. Somewhat related to the previous comment, did the authors observe a difference in miR-206 levels in young vs. older adults or trained vs. untrained participants in the METAMEX data that could explain the significantly elevated FAPs NES score figure 2D?

[Author action #2]

Thank you for raising this critical point. In terms of the miR-206 response to exercise, METAMEX lacks microRNA data and, therefore, we cannot determine the miR-206 response to resistance exercise versus immobilization. As an alternative, we conducted systematic literature review to summarize the change in miR-206 level in elderly skeletal muscle in response to exercise training. According to the systematic search, only one study to date assessed miR-206 level in elderly skeletal muscle before and after an acute bout of resistance exercise¹. The study demonstrated that pri-miR-206 levels in elderly skeletal muscle was significantly increased after an acute bout of exercise¹, thereby supporting our hypothesis that exercise may counteract the FAP adipogenesis program induced by knock out of miR-206.

To further support the effect of exercise and immobilization on miR-206, we accessed the METAMEX dataset to determine whether chronic resistance exercise and immobilization differentially change the expression level of miR-206 target genes. According to target gene prediction via miRTarBase², miR-206 directly targets 50 genes. Principal component analysis revealed that the transcriptomic response of miR-206 target genes was significantly different across exercise and immobilization protocols (**Figure S2**). Together, these new findings support the proposed mechanism that exercise regulates FAP adipogenesis genes that are regulated by miR-206. We have revised RESULTS section as follows:

Page 8 / Line 195-198 in RESULTS:

“A previous study showed that an acute bout of exercise increases pri-miR-206, the initial product of miR-206 gene transcription, in elderly skeletal muscle (Drummond et al., 2008). These results raised for us the novel hypothesis that exercise counteracts the FAP adipogenesis program induced by knockout of miR-206.”

Page 9 / Line 217-219 in RESULTS:

“Further supporting the regulatory effects of physical activity level on miR-206-dependent FAP adipogenesis, exercise and immobilization differentially regulated miR-206 target genes (Figure S2).”

Figure S2. miR-206 enrichment analysis for the transcriptomic response to resistance exercise and immobilization protocols in elderly individuals

Reference

1. Drummond MJ, et al. Aging differentially affects human skeletal muscle microRNA expression at rest and after an anabolic stimulus of resistance exercise and essential amino acids. *Am J Physiol Endocrinol Metab* 2008;295:E1333-1340.
2. Huang H-Y, et al. miRTarBase 2020: updates to the experimentally validated microRNA–target interaction database. *Nucleic acids research* 2020;48:D148-D154.

[Comment #3]

3. *The authors pick out Sparc11 as one of the 5 leading edge genes that were significantly changed across exercise and immobilization protocols. The potential importance of Sparc11 in IMAT formation has been shown already, making it a logical molecule of interest, but as the authors point out, Sparc11 is also involved in many physiological processes other than adipogenesis. At most, the identification of Sparc11 in this study is confirmatory.*

[Author action #3]

We received similar comments from Reviewer #2 (see **Comments #9** and **#12**). In the light of these comments and given a comprehensive explanation for *Sparc11* may disrupt the paper flow, we have shortened these sentences (**Page 10 / Line 233-235**).

[Comment #4]

4. *Using the other 4 leading edge genes, the authors then take a data-driven approach to construct a regulatory gene network that they claim to play a role in FAPs conversion to adipocytes. While this is not an unreasonable approach to develop new hypotheses and candidate pathways/networks linked with those 4 molecules, the risk is in inappropriately ascribing uncovered networks to a specific physiological process (i.e, adipogenesis) when those 4 genes are perhaps just red herrings. Nevertheless, pgc1a was identified as a hub gene and proposed to be a central regulator of FAPs adipogenesis. Despite extensive in silico and in situ modeling to confirm/validate that the 5 leading edge genes co-regulate with the pgc1a node with exercise and immobilization, it is difficult to ignore the possibility that 4-5 genes identified from the 161 differentially expressed mouse genes in response to a single miRNA manipulation may not translate well into the human situation.*

[Author action #4]

This is a great point. As described in **Author action #2**, we have now added new studies demonstrating that exercise and immobilization protocols differentially regulated human MME⁺ FAP signature genes that have high adipogenic potential (**Figure S3**).

Reviewer #2

[Comment #5]

This manuscript by Iijima et al aims to leverage existing transcriptomic data sets to unveil mechanistic drivers of FAP adipogenesis in the aged which are reversed, at least in part, by exercise intervention. The authors conducted analysis on several data sets of human studies as well as murine investigations which together elucidated PGC1a as a critical hub which is upregulated by exercise and may downregulate adipogenesis in the elderly. Using in silico analysis, they go on to show that PGC1a-mediated regulation of mitochondrial fatty acid oxidation may be a key inhibitor of FAP adipogenesis and thus fat accumulation in muscle. The authors should be complimented for sophisticated analysis using existing data sets and

well-written manuscript to contribute a novel finding regarding the important topic regarding skeletal muscle health in the aged. However, some concerns reduce enthusiasm for the work put forth, as detailed in a point-by-point fashion below.

Major

1. Although the authors describe additional filtering strategies for low read counts (which is not defined) and signal intensity, there is some reservation whether different read depths were corrected for to make the data comparable among the different studies and datasets. It would be helpful to have a bioinformatic statistician review the conducted analyses given their complex nature.

[Author action #5]

Thank you for your comments. Regarding the filtering for RNA-seq data, we used the filterByExpr function with default parameter for low count data (min.count = 10). We have revised the Methods to explain the filtering strategies (see **Page 23 / Line 556-557**).

Regarding the comment pertaining to different read depths, the corresponding author of this paper is a bioinformatics expert, and he has confirmed that the library size in each data set does not influence the log fold change value, and therefore, no batch effects across different studies are expected from the different read depths. There is also a Bioconductor help page discussing this topic, which shows that library size does not influence the fold change value (<https://support.bioconductor.org/p/42482/>).

[Comment #6]

2. It is appreciated that the authors communicated a clear story regarding exercise-mediated regulation of insulin sensitivity and FAP adipogenesis. However, in that focus there seems to be a missed opportunity to communicate other potentially interesting findings which would be of use to the field. In Figure 1, the authors immediately filtered for insulin sensitivity-related genes prior to conducting PCA analysis. Could the authors conduct exploratory PCA and enrichment analyses on the most up- and down-regulated DEGs without filtering for insulin-sensitivity genes? Not only would it be interesting to determine if the grouping relationships on the PCA are still evident even when not selecting for specific genes, but also would be interesting to see if the most differentially regulated enrichment pathways are metabolism-related, thus giving further support to the current study.

[Author action #6]

To address this comment, we have added a series of analyses, which are now summarized in **Figure S1**. First, we performed PCA for all 7,468 genes in the current data set, including insulin-sensitive genes, as suggested. Even with these additional genes, PCA revealed a clear segregation in the transcriptomic response of 7,468 genes when comparing immobilization and resistance exercise protocol (**Figure S1A**). Moreover, consistent with the original analysis that included insulin-sensitive genes, the principal component score, which represents overall gene expression, was significantly different across resistance exercise and immobilization groups (**Figure S1B**).

We next sought to determine the most differentially regulated pathways across the immobilization and resistance exercise protocol. For this purpose, using a loading score of each gene contributing to the principal component score as an input, we performed gene set enrichment analysis for the Wiki pathway used as a gene set (gene lists were accessed via Enrichr software). The most differentially regulated pathways were related to mitochondrial integrity such as “*Electron Transport Chain*”, “*Mitochondrial complex I assembly model OXPHOS system*” (**Figure S1C**), all of which are metabolism-related pathways, as Reviewer #2 suspected. These results support the mechanistic link between exercise, *Pgc-1a*, and mitochondrial fatty acid oxidation. We have revised the RESULTS as follows:

Page 7 / Line 155-158 in RESULTS:

“Of note, even when we considered all 7,468 genes, resistance exercise and immobilization were found to exert opposing effects on metabolism-related pathways (Figure S1). These findings are consistent with results including the insulin sensitivity genes, as shown above.

Figure S1. Overall transcriptomic response of 7,468 genes to resistance exercise and immobilization protocols in elderly individuals

[Comment #7]

3. Similar to point 2, could the authors provide the other differentially regulated pathways identified in Fig. 5 B and E as supplemental information? Although a straightforward story is appreciated, it seems like there is data that could be advantageous to readers that is currently not communicated in its current form.

[Author action #7]

We have now provided all raw data as supplementary material (see Source of data).

[Comment #8]

4. A fair degree of weight is placed upon the FAP adipogenesis-related dataset provided by Wosczyzna et al 2021 in which IMAT is regulated by miR-2016 (mimicry or inhibition) which the authors conducted analysis on to identify a list of 161 genes that "drive adipocytic differentiation of FAPs." This potentially problematic since these DEGs are only those regulated under manipulation of miR-206 (to my understanding), and thus the list is likely biased and limited since other genes surely may play a role in FAP adipogenesis but were simply not captured by manipulation of miR-206. This is critical since this DEG list is the foundation for their study. The authors should consider providing the list of 161 DEGs as a supplement, identify genes or

conduct enrichment pathway analysis which instill further confidence in the list, and provide discussion of this limitation.

[Author action #8]

Thank you for raising these important suggestions. We received a similar comment from Reviewer #1 (see **Comment #2**). We have now provided the list for 161 differentially expressed genes across wild type and miR-206 KO mice in “**Source of data**”, and we have added new analyses investigating the biological role of these genes at system level. Wiki pathway enrichment analysis for 161 genes revealed that adipogenesis-related pathways, “*Adipogenesis*” and “*PPAR signaling pathway*”, were the top 2 significant pathways significantly regulated by miR-206 KO (see **Figure 2B**). We have explained these results accordingly (see **Page 8-9 / Line 198-200 in RESULTS**). Further, we have acknowledged the limitation of the use of these genes as follows:

Page 20 / Line 485-489 in DISCUSSION:

“Finally, FAP adipogenesis genes were defined according to archived RNA-seq data from mouse skeletal muscle under manipulation of miR-206 (Wosczyzna et al., 2021). Although growing evidence indicates that murine FAPs have a transcriptomic profile similar to human FAPs (Fitzgerald et al., 2023), the proposed mechanisms may not completely capture the dynamics of FAP adipogenesis observed in elderly skeletal muscle.”

Figure 2B. Top five Wiki pathways significantly enriched to 161 differentially expressed genes associated with FAP adipogenesis.

[Comment #9]

5. *Of the 5 leading-edge genes identified (first described in Fig. 2D; Bgn, col5a2, Coll4a1, Sparcl1, and Timp1), it is understood why the authors focused on Sparcl1 given its suggested role in adipogenesis. Collectively, however, this list seems to indicate extracellular matrix remodeling as the general function of these leading-edge genes. This would logically make sense if exercise mediates upregulation of ECM remodeling which then in turn would prevent FAP adipogenic programs. Do the authors think activation of ECM programming diametrically opposes adipogenesis?*

[Author action #9]

Thank you for this excellent question. As the reviewer has pointed out, we cannot exclude the possibility that matrix remodeling mediates the effects of exercise on the regulation of the FAP adipogenesis program. Indeed, FAP adipogenesis induced by miR-206 deletion was accompanied by significant changes in genes associated with focal adhesion, a process required for matrix remodeling (**Figure 2B in Author action #10**). There is evidence that FAP differentiation is regulated by biophysical properties of the extracellular matrix¹. On the other hand, dysregulation of FAP differentiation also leads to fibrosis, a leading cause of matrix stiffening². This bidirectional interaction between FAPs and their surrounding matrix, a phenomenon known as dynamic reciprocity³, is an area of interest for future studies. We have discussed these points as follows:

Page 19-20 / Line 467-477 in DISCUSSION:

“It is noteworthy that FAP adipogenesis induced by miR-206 deletion was accompanied by extracellular matrix remodeling, as documented by significant changes in focal adhesion signaling. Accordingly, the five leading edge genes that were predominantly regulated by exercise and immobilization code for extracellular matrix-related proteins. As such, it is possible that matrix remodeling mediates the effects of exercise on the regulation of the FAP adipogenesis program. Indeed, there is evidence that FAP differentiation is regulated by biophysical properties of extracellular matrix(Loomis et al., 2022). Dysregulation of FAP differentiation also leads to fibrosis, a leading cause of matrix stiffening(Loomis & Smith, 2023). This bidirectional interaction between FAPs and their surrounding matrix, a phenomenon known as dynamic reciprocity(Bissell, Hall & Parry, 1982), is an area of interest for future studies.”

Reference

1. Loomis T, et al. Matrix stiffness and architecture drive fibro-adipogenic progenitors' activation into myofibroblasts. *Sci Rep* 2022;12(1):13582.
2. Loomis T, et al. Thrown for a Loop: Fibro-Adipogenic Progenitors in Skeletal Muscle Fibrosis. *Am J Physiol Cell Physiol* 2023.
3. Bissell MJ, et al. How does the extracellular matrix direct gene expression? *J Theor Biol* 1982;99:31-68.

[Comment #10]

6. Could the authors provide additional information on "module 8" in Figure 3 given that one of the 5 identified leading-edge genes (Col5a2) was enriched in this module and may provide further insight to the mechanism by which exercise regulates FAP adipogenesis?

[Author action #10]

To address this comment, we have performed Wiki pathway enrichment analysis for genes in module 8, which revealed that the top pathway identified was “Focal adhesion”. Importantly, adipogenesis-related pathways were not significantly associated with module 8. We have provided these results as follows:

Page 12 / Line 275-279 in RESULTS:

“Of note, genes in module 8, another module that included the leading edge gene (Col5a2), were predominantly associated with “Focal adhesion”, but not adipogenesis-related pathways. The distinct pathway enrichment of modules 2 and 8 implies a unique biological role of module 2 in regulating adipogenesis in skeletal muscle.”

[Comment #11]

Minor

1. Could the authors indicate study exclusion criteria in the manuscript text to accompany/expand upon rationale for the information provided in Figure 1A?

[Author action #11]

We have added exclusion criteria with the rationale as follows:

Page 6 / Line 136-140 in RESULTS:

“Studies with non-healthy adults (e.g., diabetes and/or metabolic syndrome), young or middle-aged adults, non-resistance exercise (e.g., aerobic exercise), adults who performed life-long activity, and adults who participated in athlete-level physical activity were excluded in order to minimize possible confounding effects on the transcriptomic response to exercise protocol.”

[Comment #12]

2. *Quite a bit of discussion is added throughout the results section, which is a stylistic choice and is helpful at times, but distracting at others. The authors should consider shortening lines 228-237 and 280-289 which are not necessary to logically follow the respective subsequent results sections.*

[Author action #12]

We have shortened the lines 228-237 and 280-289 as suggested.

[Comment #13]

3. *The authors should define WGCNA in line 254.*

[Author action #13]

We have defined WGCNA as suggested (see **Page 11 / Line 253-255**).

[Comment #14]

4. *Throughout the manuscript, the color blue notates immobilization and the color yellow notates exercise intervention. Thus, using this same color scheme (blue and yellow) to notate sex (male and female) in Figure 1B is confusing.*

[Author action #14]

Thank you for this suggestion. We have changed the color scheme in **Figure 1B**, using green for males and orange for females.

[Comment #15]

5. *Figure 3E is confusing in its current form and doesn't seem to add much to the information already provided in Figure 3D. The authors should consider editing or removing.*

[Author action #15]

We have deleted Figure 3E as suggested (see **Figure 3**).

Dear Dr Iijima,

Re: JP-RP-2023-285349R1 "Network-based systematic dissection of exercise-induced inhibition of myosteatosis in older individuals" by Hiroataka Iijima, Fabrisia Ambrosio, and Yusuke Matsui

Thank you for submitting your manuscript to The Journal of Physiology. It has been assessed by a Reviewing Editor and by 3 expert referees and we are pleased to tell you that it is acceptable for publication following satisfactory revision.

REVISION CHECKLIST:

Please upload two versions of your manuscript text: one with all relevant changes highlighted and one clean version with no changes tracked. The manuscript file should include all tables and figure legends, but each figure/graph should be uploaded as separate, high-resolution files. The journal is now integrated with Wiley's Image Checking service. For further details, see: <https://www.wiley.com/en-us/network/publishing/research-publishing/trending-stories/upholding-image-integrity-wileys-image-screening-service>.

We look forward to receiving your revised submission.

Yours sincerely,

Paul Greenhaff
Senior Editor
The Journal of Physiology

REQUIRED ITEMS

-Your paper contains Supporting Information of a type that we no longer publish. Any information essential to an understanding of the paper must be included as part of the main manuscript and figures. The only Supporting Information that we publish are video and audio, 3D structures, program codes and large data files. Your revised paper will be returned to you if it does not adhere to our Supporting Information Guidelines

-Papers must comply with the Statistics Policy https://jp.msubmit.net/cgi-bin/main.plex?form_type=display_requirements#statistics

In summary:

-If $n \leq 30$, all data points must be plotted in the figure in a way that reveals their range and distribution. A bar graph with data points overlaid, a box and whisker plot or a violin plot (preferably with data points included) are acceptable formats.

-If $n > 30$, then the entire raw dataset must be made available either as supporting information, or hosted on a not-for-profit repository e.g. FigShare, with access details provided in the manuscript.

- 'n' clearly defined (e.g. x cells from y slices in z animals) in the Methods. Authors should be mindful of pseudoreplication.

-All relevant 'n' values must be clearly stated in the main text, figures and tables

-The most appropriate summary statistic (e.g. mean or median and standard deviation) must be used. Standard Error of the Mean (SEM) alone is not permitted, unless justified and presented alongside confidence intervals.

-Exact p values must be stated. Authors must not use 'greater than' or 'less than'. Exact p values must be stated to three significant figures even when 'no statistical significance' is claimed.

EDITOR COMMENTS

Reviewing Editor:

Thank you for submitting your work to The Journal of Physiology and for providing a thorough revision of your manuscript. All major concerns raised by the two reviewers have been adequately addressed. However, The Journal does not currently support the inclusion of supplementary figures, and thus, the authors need to determine whether the supplementary figures should be included as figures in the manuscript or removed entirely. In addition, there is concern regarding the statistical approaches given the small sample size in this study. The authors need to adequately address the comments provided by the statistical editor before a final decision on acceptance can be made.

Senior Editor:

Thank you for your revised manuscript that has been considered by the same two expert reviewers and RE that considered the original submission, as well as a statistical editor. The original reviewers believe the manuscript is acceptable for publication. However, the statistical editor has raised concerns of potential low n's and inappropriate statistics being employed on a number of occasions. Although the manuscript is deemed to be provisionally acceptable, these concerns obviously need to be addressed by the authors if the manuscript is to be considered further. I therefore strongly encourage the authors to provide a comprehensive response in their revisions. The study is appreciated for sure. The question is around the robustness and generalisability of the data, which the authors need to address.

REFEREE COMMENTS

Referee #1:

The authors were thoughtful and genuine in their responses to my initial queries and comments.

Referee #2:

The authors have satisfied my previous concerns through a thorough revision which has strengthened the manuscript. I have no additional comments.

Referee #3:

I share the senior editor's concerns regarding the extremely small sample size (i.e., $n=2/3$ being compared to $n=5$), which severely limits the robustness and generalisability of these data. The manuscript provides no detail on how the sample size was determined and whether a power analysis was conducted to ensure that the study was adequately powered to detect the expected effects. Inferences from parametric methods are very inaccurate for small sample sizes unless normality and equal variance are very nearly true. I would advise the authors to use non-parametric analyses here (Wilcoxon and Mann-Whitney for paired and unpaired tests, respectively), as these hold up better with small sample sizes. Similarly, conducting PCA analyses on such small samples is not recommended, and not enough detail was provided to determine whether the PCA analysis was appropriate here. Specifically, the methods should include information on data pre-processing (e.g., scaling and centering of data), how the number of principal components to retain was determined, and how the assumptions of PCA were checked (and the outcomes of these) e.g., Bartlett's Test of Sphericity and the Kaiser-Meyer-Olkin (KMO) Measure of Sampling Adequacy).

The methods did not describe where/when each statistical test was used, and why. For instance, I could not where the Fisher's exact test was used, or why this was deemed appropriate. There is limited detail provided on assumptions and diagnostics checks - the manuscript mentions checking the residuals vs. fitted values for the regression model but it would be helpful to know if other assumptions of the linear regression were checked, such as homoscedasticity).

The decision not to include confounders (e.g., mean body mass) in the analysis was explained by the small sample size. However, it would be helpful to know how this might affect the validity of the results and what steps were taken to minimize the impact of this decision. I'm not familiar with the physiological underpinnings, but it seems possible that a confounder such as mean body mass could explain much of the variance/difference observed.

The manuscript mentions using a complete case analysis for handling missing data. It would be helpful to know how much data were missing, whether the potential impact of this method on the results was considered, and whether alternatives

were considered e.g., multiple imputation.

Overall, I would be very hesitant to publish a paper making conclusions such as the one below based on such small samples and limited methodological description.

"The current study suggests that physical activity reduces fat in skeletal muscle via upregulation of Pgc-1 α -mediated mitochondrial fatty acid oxidation and subsequent inhibition of FAP adipogenesis"

END OF COMMENTS

1st Confidential Review

27-Sep-2023

We are grateful to the editors for providing additional feedback, which we believe has made the work even stronger. The point-to-point reply to the statistical editor's comments is found below.

Reviewer #1

[Comment #1]

I share the senior editor's concerns regarding the extremely small sample size (i.e., $n=2/3$ being compared to $n=5$), which severely limits the robustness and generalisability of these data. The manuscript provides no detail on how the sample size was determined and whether a power analysis was conducted to ensure that the study was adequately powered to detect the expected effects. Inferences from parametric methods are very inaccurate for small sample sizes unless normality and equal variance are very nearly true. I would advise the authors to use non-parametric analyses here (Wilcoxon and Mann-Whitney for paired and unpaired tests, respectively), as these hold up better with small sample sizes. Similarly, conducting PCA analyses on such small samples is not recommended, and not enough detail was provided to determine whether the PCA analysis was appropriate here. Specifically, the methods should include information on data pre-processing (e.g., scaling and centering of data), how the number of principal components to retain was determined, and how the assumptions of PCA were checked (and the outcomes of these) e.g., Bartlett's Test of Sphericity and the Kaiser-Meyer-Olkin (KMO) Measure of Sampling Adequacy).

[Author action #1]

We are very grateful for this critical feedback. We recognize that the findings of this study are exploratory in nature and based on a small number of archived datasets available. We have revised the text as follows:

Page 20 / Line 489-490 in DISCUSSION:

“First, findings of this study are exploratory in nature and are based on a small number of currently available datasets that employ an immobilization model.”

Page 24 / Line 644-645 in MATERIALS and METHODS:

“Due to the exploratory nature of the study, the sample size was not pre-determined.”

We agree that Student t-test is not appropriate in the case of small sample sizes as ours. As an alternative, we used the non-parametric test (Mann-Whitney U-test) as suggested. The overall results are consistent with the initial analyses. We have revised the manuscript and figures accordingly (please see **Page 24 / Line 645 in MATERIALS and METHODS, Figures, and Figure captions**).

Regarding PCA for skeletal muscle transcriptomic response across exercise and immobilization protocols, we followed analytical flow of PCA as per previous studies that used MetaMEx datasets¹⁻². However, to address Editors' recommendations, we have checked structure of data matrix via correlogram, which is required to check fundamental assumptions as per previous literature³. The heatmap clearly illustrates the gene “module” structure and linearity among all 7,468 genes (**Reference Figure 1**), therefore meeting the assumptions for the linearity and suitability for data reduction. Linearity of the data was also confirmed via Pearson correlation analysis for randomly selected gene pairs (**Reference Figure 2**). In addition, the significance of data reduction was checked via PCA scree plot as documented by high proportion of variance explained by PC1 and PC2 (**Reference Figure 3**). We have acknowledged these points as follows:

Page 24 / Line 631-636 in MATERIALS and METHODS:

“Prior to analysis, we checked assumptions for linearity and suitability of data reduction via correlogram, as per a previous report(Bair et al., 2006). To further confirm the linearity, a Pearson correlation analysis was performed for randomly selected genes, which supported the linear relationship of transcriptomic

response among genes. In addition, we confirmed the significance data reduction via a PCA scree plot, which represents the amount of variance explained by each principal component.”

We understand that the main suggestion raised by the Editor was to provide additional results and/or discussion regarding the robustness and generalizability of the main findings given the small sample size in PCA. As such, we further directly assessed the robustness of PCA to small sample size by implementing a leave-one-out method. A randomly selected dataset was removed from the PCA, and this process was repeated for each of the datasets. The results still revealed a clear segregation in the transcriptomic response when comparing immobilization and resistance exercise protocols (**Reference Figure 4**). These new findings support the robustness of our findings to small number of datasets. We have acknowledged these points as follows:

Page 24 / Line 638-640 in MATERIALS and METHODS:

“We confirmed similar PCA results of original analysis, as documented by a clear segregation in the transcriptomic response when comparing exercise and immobilization protocols.”

Reference Figure 1. Correlogram illustrating a gene module structure of muscle transcriptomic response to exercise and immobilization protocols (7,468 genes in total). Color indicates Spearman correlation coefficient (red: -1, blue: 1).

Reference Figure 2. Scatter plot with fitting line for transcriptomic response of randomly selected gene pair (2x4 genes). Each figure indicates a linear relationship between randomly selected gene pair.

Reference Figure 3. Scree plot of PCA demonstrating that significant data reduction of overall transcriptomic response was confirmed by PCA, as documented by high proportion of variance explained by PC1 and PC2.

Reference Figure 4. Biplot and scree plot of PCA after randomly removed one dataset (leave-one-out; n=6 in each plot), demonstrating a clear segregation between exercise and immobilization protocols and significant data reduction of overall transcriptomic response across the simulations.

Reference

1. Pillon NJ, et al. Transcriptomic profiling of skeletal muscle adaptations to exercise and inactivity. *Nat Commun* 2020;11:470.
2. Malenfant P, et al. Fat content in individual muscle fibers of lean and obese subjects. *Int J Obes Relat Metab Disord* 2000;25:1316-1321.
3. Bair E, et al. Prediction by supervised principal components. *Journal of the American Statistical Association* 2006;101:119-137.

[Comment #2]

The methods did not describe where/when each statistical test was used, and why. For instance, I could not where the Fisher's exact test was used, or why this was deemed appropriate. There is limited detail provided on assumptions and diagnostics checks - the manuscript mentions checking the residuals vs. fitted values for the regression model but it would be helpful to know if other assumptions of the linear regression were checked, such as homoscedasticity).

[Author action #2]

We apologize for the lack of clarity. The current study did not use the Fisher's exact test and we deleted it from the MATERIALS and METHODS section. Regarding the linear regression analysis, we have checked assumptions using a Shapiro-Wilk test and residuals vs. fitted values. We have acknowledged these points as follows:

Page 24 / Line 645-648 in MATERIALS and METHODS:

"In the linear regression analysis, we checked the (1) normality of residuals using a Shapiro-Wilk test, (2) homogeneity of the variances as well as linearity by comparing the residuals vs. fitted values (i.e., the residuals had to be normally distributed around zero)."

[Comment #3]

The decision not to include confounders (e.g., mean body mass) in the analysis was explained by the small sample size. However, it would be helpful to know how this might affect the validity of the results and what steps were taken to minimize the impact of this decision. I'm not familiar with the physiological underpinnings, but it seems possible that a confounder such as mean body mass could explain much of the variance/difference observed.

[Author action #3]

Thank you for raising these thoughtful suggestions. Studies indicate the established association between obesity (i.e., high body mass index) and IMAT^{1,2}. As such, we cannot discount the possibility that the impact of physical activity level on FAP adipogenesis may be a function of participants' body mass. Nevertheless, at the whole study level, participants' BMIs were similar across resistance exercise and immobilization protocols, suggesting minimal likely impacts of BMI on the distinct FAP regulatory effects observed across the two protocols. We have acknowledged these points as follows:

Page 20-21 / Line 489-498 in DISCUSSION:

"First, findings of this study are exploratory in nature and are based on a small number of currently available datasets that employ an immobilization model. This may contribute to bias depending on the participants' characteristics and the methodology used in the original studies. Prior studies have indicated an association between obesity (i.e., high body mass index) and IMAT (Goodpaster et al., 2000; Malenfant et al., 2001). As such, we cannot discount the possibility that the impact of physical activity level on FAP adipogenesis may be a function of participants' body mass. Nevertheless, the averaged BMI for participants across resistance exercise and immobilization protocols was similar, indicating a likely minimal impact of BMI on the distinct FAP regulatory effects observed across the two protocols."

Reference

1. Goodpaster BH, et al. Intramuscular lipid content is increased in obesity and decreased by weight loss. *Metabolism* 2000;49:467-472.
2. Malenfant P, et al. Fat content in individual muscle fibers of lean and obese subjects. *Int J Obes Relat Metab Disord* 2000;25:1316-1321.

[Comment #4]

The manuscript mentions using a complete case analysis for handling missing data. It would be helpful to know how much data were missing, whether the potential impact of this method on the results was considered, and whether alternatives were considered e.g., multiple imputation.

[Author action #4]

Thank you for this critical reminder. All of the missing data was from the MetaMEx database (i.e., RNA-seq and/or microarray data sets). Each dataset has missing value for genes, which is attributed to low counts or low signal intensity. From a statistical point of view, genes with consistently low counts or low signal intensity are unlikely to be assessed as differentially expressed because low counts or low signal intensity do not provide enough statistical evidence for a reliable judgement to be made¹. Such genes can therefore be removed from the analysis without any loss of information. As such, we did not consider multiple imputation. We have now explained these points as follows:

Page 24 / Line 648-651 in MATERIALS and METHODS:

“We conducted a complete-case analysis in the case of missing data for RNA-seq and/or microarray (i.e., genes under detection). As the missing data with low counts or low signal intensity do not provide enough statistical evidence for a reliable judgement to be made, we did not consider multiple imputation.”

Reference

1. Chen Y, et al. From reads to genes to pathways: differential expression analysis of RNA-Seq experiments using Rsubread and the edgeR quasi-likelihood pipeline. *F1000Res* 2016;5:1438.

[Comment #5]

Overall, I would be very hesitant to publish a paper making conclusions such as the one below based on such small samples and limited methodological description.

“The current study suggests that physical activity reduces fat in skeletal muscle via upregulation of Pgc-1 α -mediated mitochondrial fatty acid oxidation and subsequent inhibition of FAP adipogenesis”

[Author action #5]

To address this issue, we have emphasized the exploratory nature of the study findings by revising the conclusion as follows:

Page 2 / Line 46-48 in KEY POINTS SUMMARY:

“Together, the findings of the current study suggest a novel hypothesis that physical activity reduces fat in skeletal muscle via upregulation of Pgc-1 α -mediated mitochondrial fatty acid oxidation and subsequent inhibition of FAP adipogenesis.”

Dear Dr Iijima,

Re: JP-RP-2023-285349R2 "Network-based systematic dissection of exercise-induced inhibition of myosteatosis in older individuals" by Hirotaka Iijima, Fabrisia Ambrosio, and Yusuke Matsui

We are pleased to tell you that your paper has been accepted for publication in The Journal of Physiology.

Authors should note that it is too late at this point to offer corrections prior to proofing. The accepted version will be published online, ahead of the copy edited and typeset version being made available. Major corrections at proof stage, such as changes to figures, will be referred to the Editors for approval before they can be incorporated. Only minor changes, such as to style and consistency, should be made at proof stage. Changes that need to be made after proof stage will usually require a formal correction notice.

Yours sincerely,

Paul Greenhaff
Senior Editor
The Journal of Physiology

P.S. - You can help your research get the attention it deserves! Check out Wiley's free Promotion Guide for best-practice recommendations for promoting your work at www.wileyauthors.com/eeo/guide. You can learn more about Wiley Editing Services which offers professional video, design, and writing services to create shareable video abstracts, infographics, conference posters, lay summaries, and research news stories for your research at www.wileyauthors.com/eeo/promotion.

IMPORTANT NOTICE ABOUT OPEN ACCESS: To assist authors whose funding agencies mandate public access to published research findings sooner than 12 months after publication, The Journal of Physiology allows authors to pay an Open Access (OA) fee to have their papers made freely available immediately on publication.

You can check if your funder or institution has a Wiley Open Access Account here: <https://authorservices.wiley.com/author-resources/Journal-Authors/licensing-and-open-access/open-access/author-compliance-tool.html>.

EDITOR COMMENTS

Reviewing Editor:

Thank you for thorough reassessment of the statistical approaches used due to the limited number of datasets available for this study. All the revisions have been adequately addressed, and I would like to congratulate the authors on the completion of an excellent study. Thank you for submitting your work to the special issue on the Physiology of Ageing Skeletal Muscle and the Protective Effects of Exercise in The Journal of Physiology.

Senior Editor:

Thank you for addressing the comments raised by the Statistical Editor and revising the manuscript accordingly. The manuscript has been improved as a result.

REFEREE COMMENTS

Referee #3:

Thank you for addressing my points clearly and robustly, and for undertaking the additional analyses. I am pleased to see you have now clearly emphasised the exploratory nature of this work. I am satisfied with the changes made and additional analyses/explanations provided.

2nd Confidential Review

07-Nov-2023